# Microfluidic chips provide visual access to in situ soil ecology

Paola Micaela Mafla-Endara [1,2✉], Carlos Arellano-Caicedo[1], Kristin Aleklett [1,3], Milda Pucetaite [1], Pelle Ohlsson [4] & Edith C. Hammer [1,2✉]

Microbes govern most soil functions, but investigation of these processes at the scale of their cells has been difficult to accomplish. Here we incubate microfabricated, transparent 'soil chips' with soil, or bury them directly in the field. Both soil microbes and minerals enter the chips, which enables us to investigate diverse community interdependences, such as inter-kingdom and food-web interactions, and feedbacks between microbes and the pore space microstructures. The presence of hyphae ('fungal highways') strongly and frequently increases the dispersal range and abundance of water-dwelling organisms such as bacteria and protists across air pockets. Physical forces such as water movements, but also organisms and especially fungi form new microhabitats by altering the pore space architecture and distribution of soil minerals in the chip. We show that soil chips hold a large potential for studying *in-situ* microbial interactions and soil functions, and to interconnect field microbial ecology with laboratory experiments.

[1] Department of Biology, Lund University, Lund, Sweden. [2] Centre for Environmental and Climate Science, CEC, Lund University, Lund, Sweden. [3] Department of Plant Protection Biology, Swedish University of Agricultural Sciences, Alnarp, Sweden. [4] Department of Biomedical Engineering, Lund University, Lund, Sweden. ✉email: paola_micaela.mafla_endara@biol.lu.se; edith.hammer@biol.lu.se

Soil microorganisms are essential for nutrient cycling, soil aggregation, and regulation of soil carbon storage. Their home, the soil, consists of matrices of mineral particles that harbor partly interconnected pores. These pore spaces are full of contrasting microhabitats of varying sizes and chemical conditions, in which microbes live, and which they constantly re-shape[1,2]. The physical microstructure of soil is a strong determinant of soil functions and ecological interactions[3,4]. It makes the soil habitat unique, extraordinarily species rich[5], and allows for the accumulation of organic matter despite the presence of many substrate-limited microorganisms[6,7]. The arrangement and structure of soil aggregates and pore spaces define the connectivity between microhabitats and thus the soil microorganisms' access to or restriction from different resources (e.g., food sources, water, and oxygen) based on their dispersal potential[8,9]. The system is highly dynamic in space and time: solid particles are moved by both physical processes and biota, and additional barriers for diffusion and mass flow occur when pores dry out, inhibiting the dispersal of microorganisms such as bacteria and protists that are reliant on hydraulic connectivity for relocation[9–11].

Despite the continuous effort to understand how communities of soil microbes function and contribute to soil processes, current techniques have not been able to completely address the complexity of their spatiotemporal organization at the microscale[1,12–14]. For organisms living inside the minuscule, solid soil microstructures, it is expected that most of the cell-to-cell interactions take place over short distances, generally no more than a few tens of micrometers[15]. Identifying and studying these microhabitats is challenging as their spatial organization is easily destroyed and lost during sample processing[13,16]. As a result, the capacity for addressing fundamental knowledge gaps in the field of soil science has been limited, including the impact of spatial microstructures on biogeochemical processes like nutrient cycling, feedbacks between microbes and soil physical processes, inter-kingdom interactions, and biodiversity-function relationships[17–19]. We have now developed a microfluidic system that allows us to experimentally address these questions in situ and with higher spatial control than previously possible. Natural microbial communities can be inoculated into these systems, selected only by their ability to pass through the entrance of limited size.

Microfluidic chips have already demonstrated their usefulness in controlling and shaping micro-environments for the study of cell-to-cell interactions, and revolutionized biomedical research with, e.g., organ-on-a-chip devices[20]. Even within soil science and microbial ecology, chips have been used to address important questions[21,22] such as how to increase the number of culturable bacteria from the environment[23], how bacteria spatially organize in a pore space along chemical gradients[24], and how intracellular signals propagate in fungal networks[25]. In most cases, chips have been inoculated with one or two microbial species at a time, exposed to very controlled spatial and/or chemical conditions. In this study, we used a whole-soil inoculum for microfluidic chips to investigate interactions within multi-species microbial communities including physical soil components. We studied the early microbial colonization of the chip's pristine, soil-like habitat by (I) burying it directly into the soil habitat in the field, or (II) inoculating it with soil and incubating it in the laboratory. We expect the first approach to allow us to study conditions most closely resembling those in nature, while the second approach allows us to follow processes at higher control and over time. We asked whether the dispersal capability of three functional microbial groups—fungi, bacteria, and protists—into a pristine pore space environment is influenced by pore space characteristics such as their geometric shape and chemical conditions, and by interactions with other microbes. We further examined how the microhabitats themselves are affected by abiotic and biotic

factors such as drying and rewetting of the soil, and by the microorganisms themselves.

The chip design contained different experimental sections with distinct geometrical patterns[26] (Supplementary Fig. 1a, Sections A–E), which we used to address the following specific questions: (a) How is microbial dispersal influenced by the pore spaces being filled with air, water, or nutrient medium? (b) How does pore space geometry affect microbial dispersal, such as channels angled in zigzag patterns, forcing the microbes to navigate through increasingly sharper turns? (c) Are bacteria and protists influenced in their dispersal capabilities to new pore spaces by the presence of a fungal hypha? and (d) How does drying and rewetting soil, and the moving and growing microorganisms, affect the spatial arrangement of the chips' pore space?

In summary, our experiments show that water and nutrient conditions mainly affect water-dwelling organism groups of bacteria and protists, while the shape of the microstructures has an effect on fungal dispersal. Fungal hyphae strongly enhance the colonization success for both bacteria and protists in an initially dry pore space via increased pore wetting. The chips also reveal spatiotemporal changes of microhabitats: hyphae both open up new passages in the pore space system and block them for both organisms and abiotic soil components. Water movements, triggered by drying and rewetting the soil, lead to the development of preferential water pathways that differentiate microhabitats further.

## Results

All major groups of soil microorganisms (bacteria, fungi, protists), as well as invertebrates such as nematodes and micro-arthropods, colonized the chips and explored their internal structures, both when the chips were incorporated into soil (Expt. 1) and when they were incubated with soil in the laboratory (Expt. 2 and 3). Soil mineral particles (Supplementary Fig. 2) and soil solution also entered the chips via water movements. The transparency of the chips allowed us to observe the primary colonization of a pristine pore space and soil microbial interactions in real time (Fig. 1): microhabitat formation (Fig. 1a–c, Supplementary Movies 1–2), interactions of fungal hyphae with other soil organisms and components (Fig. 1c–h, Supplementary Movies 3–7), and microbial food-web interactions (Fig. 1g–i, Supplementary Movies 7 and 8).

**Pore space geometrical characteristics and filling.** To assess the influence of growth medium conditions in the pore spaces on the dispersal of soil microbes, we filled the chips with either malt extract medium, water, or left them empty prior to incubation in or with soil. Bacteria and protists colonized the chips filled with liquid (malt extract medium or water) to a much larger extent than air-dry ones in-situ (Expt. 1, Fig. 2b, c; $F = 4.0$, $p = 0.037$ for bacteria and $F = 3.63$, $p = 0.047$ for protists; $n = 3$, DF = 8), and in the laboratory-controlled setups, both liquid-filled chips were colonized to a significantly larger extent by bacteria and protists than in the air-filled chips (Expt. 2, Fig. 2e, f; $F = 1075$, bacteria; $F = 157$, protists; $p < 0.0001$ for both; $n = 24$ (12 channels × 2 chips except for the water treatment where one chip failed), DF = 8). In contrast, fungal hyphae showed variable results, with strong or weak growth without a consistent effect of chip filling in both the field- and lab-incubated settings (Fig. 2a, d). The lab-incubated chips (Expt. 2) enabled us to investigate the colonization of pore spaces in a time-resolved manner (Supplementary Fig. 3; Supplementary Movies 9–11). The organisms entered the liquid-filled chip from within hours (bacteria) to days (fungi, protists and nematodes). Generally, cell numbers in the malt treatment were immediately higher than in the water treatment

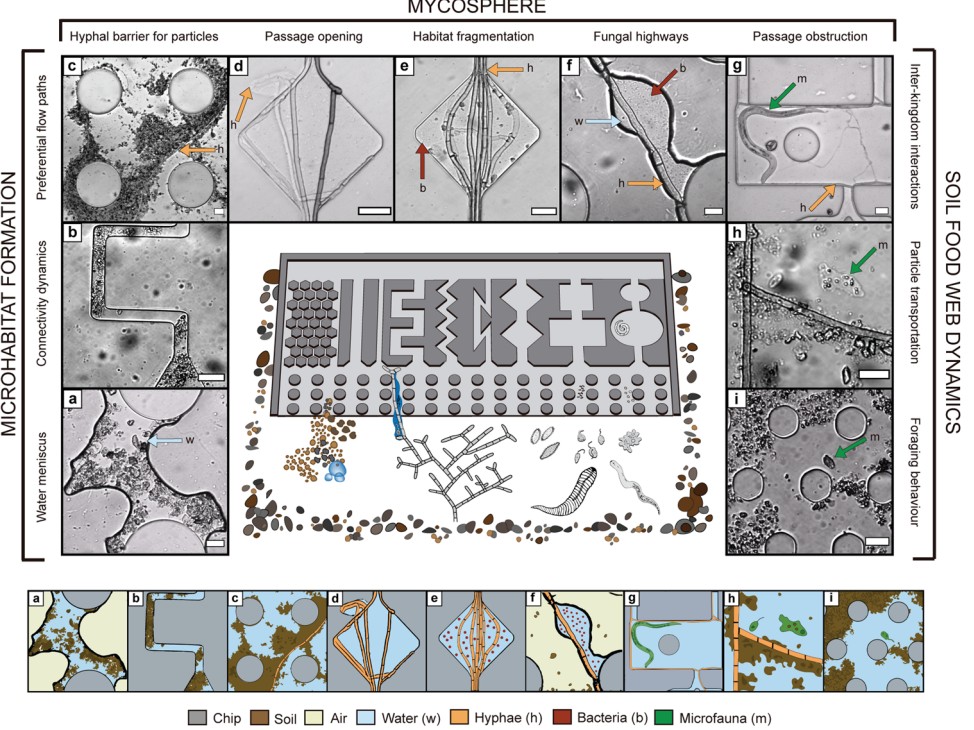

**Fig. 1 Soil organisms and processes recorded in micro-engineered soil chips.** The chips were colonized in situ from a natural soil when buried in the field. The inner section shows a schematic drawing of the design of the silicone and glass chip (dimensions exaggerated for clarity) and indicates the 7 μm high opening from where soil components and organisms can enter. Bright-field microscopy images are grouped into the research fields of microhabitat formation, the mycosphere, and soil food-web dynamics. A graphic legend below the chip explains content of the microscope images. **a** Water meniscus connecting soil particles and chip structures. **b** Mobile soil particles blocking connectivity of the artificial pore spaces. **c** Preferential water flow paths developing among soil particles sedimenting at the bottom of hyphal structures. **d** Passage opening by hyphae which squeezed through the borders of an artificial pore space, creating a new micropore. **e** Habitat fragmentation by hyphae clogging a pore neck. **f** Fungal highways: hypha-facilitated bacterial dispersal across air gaps. **g** Passage obstruction: a hypha blocking the entrances to a rectangle pore occupied by a nematode. **h** Particle transport: amoeba transporting ingested bacteria and particles. **i** Foraging behaviour: flagellated protozoa foraging around and in soil aggregates. **a**, **d**, **e** Derive from air-filled chips; **b**, **f**, **g**, and **i** derive from malt medium-filled chips; **c** and **h** derive from water-filled chips. Each type of observation was recorded at least 3 times. **a**, **b**, **d–i** Supported by Supplementary Movies 1–8, resp. Scale bars = 20 μm (**a–i**).

and maintained larger population sizes, especially for protists, throughout the experiment. We found a high turnover of fungal hyphae in the malt treatment (Supplementary Movie 11).

We further analysed the influence of 10 μm wide channel-shape geometries on the dispersal ability of the different soil microorganisms, both in the in situ-chips (Fig. 2a–c) and in the laboratory incubations (Fig. 2d–g). Water-dwelling organisms such as bacteria (Fig. 2b, e) and protists (Fig. 2c, f) were not affected in their dispersal by the three investigated channel shapes. Fungi preferred, at sufficient colonization, to grow through 'zigzag' channels, deviating 45° from the main growth direction in alternating 90° angles, over 'square' 90° angles alternating perpendicularly to the growth direction, or 'z'-channel angles of 135° (Fig. 2g. Expt. 3, air-filled chips; $F = 21.7$, $p > 0.0001$, $n = 24$ (12 channels per type × 2 chips); no significant effects found during Expt. 1 and 2, Fig. 2a, d).

**The influence of fungal hyphae on microbial dispersal**. To study the impact of fungal hyphae on the dispersal of other soil organisms, we used sets of air-filled chips where air pockets constitute obstacles for water-dwelling organisms in Expt. 1. After 2 months buried in the soil, those chips were no longer exclusively dry, as condensation water and soil solution had been dragged into the chip space, resulting in a patchy distribution of air and water in the pore space, and a subsequent colonization of its pore structures. We measured the dispersal and abundance of

bacteria in the repeated widenings (Fig. 3a) in initially air-filled channels colonized by hyphae compared to directly adjacent hypha-free channels. Results showed that the presence of hyphae, coming directly from the surrounding soil into the chip, facilitated bacterial dispersal deeper into the chip interior by enabling their passage across air pockets (Figs. 1f and 3b, d–f; Supplementary Movie 5), which resulted in a more than six-fold increase in bacterial abundance in channels with a hypha ($F = 45.6$, $p < 0.0001$; $n = 33 \times 4$; or paired $t$-test $p = 0.0008$, $n = 4$, DF = 3 Fig. 3b). Hyphal presence changed the pore space hydrology, as 80% of the pores containing hyphae filled up with liquid, in contrast to only 40% of the widenings without hyphae (Fig. 3c, ChiSquare 42.5, $p < 0.0001$, $n = 264$, DF = 1). Bacterial cell abundance commonly peaked near the hyphal frontier close to the tip (Fig. 3e), or near the furthest extent the water reached along the hypha (Fig. 3f).

While the chips embedded in-situ in a natural outdoor soil ecosystem experienced the most realistic environmental influences, the chips that were incubated with soil in the laboratory enabled us to follow processes in a time-resolved manner (Expt. 3, Fig. 4, Supplementary Movies 12–14, Supplementary Fig. 4). In air-filled pore spaces connected to soils with intermediate humidity, hyphae were commonly the first to enter the channels, shortly followed by water films and then bacteria (Fig. 4a). The presence of hyphae increased the water saturation of the pore spaces (between days 6 and 20; $F = 12.03$, $p < 0.0005$, see Fig. 4b), increasingly enhanced the bacterial populations colonizing the chips' pores during the first

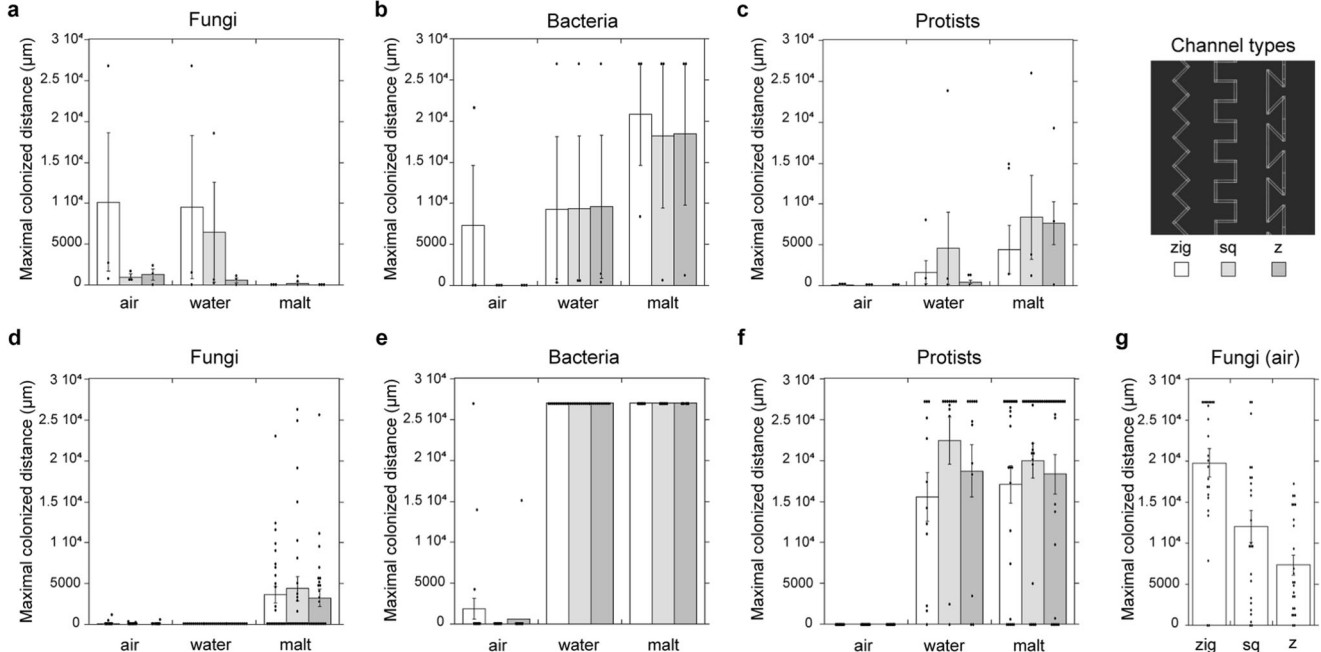

**Fig. 2 Maximum dispersal extent of different soil microbial groups.** Colonization distances of the three microbial groups, fungi, bacteria, and protists, recorded in soil chips incorporated into soil (**a**, **b**, **c**, Expt. 1, $n = 3$ chips for each condition) and incubated with soil in the laboratory (**d**, **e**, **f**, Expt. 2, $n = 2$ chips × 12 channels). **g** Fungal colonization distance in Expt. 3 (air-filled chips, $n = 2$ chips × 12 channels). The channels analysed are 10 µm wide, shaped with corners of different angles (see legend: zigzag (white bars), square (light gray bars), z-shaped (dark gray bars), under the conditions dry = air-filled, water-filled, or malt extract-filled. Error bars denote the standard error of the mean; original data points overlaying. The maximum extent of the channels was 27,000 µm and thus the maximum possible colonization extent of this experiment.

20 days by more than 8-fold (presence of hyphae*time $F = 122$, $p < 0.0001$; largest enhancement at day 12, 170-fold), and the extent of bacterial dispersal into the channels ($F = 5.7$, $p = 0.017$; Fig. 4c). At day 12, already 46% of the 108 channels were colonized by hyphae, and 10% of all channels contained bacteria growing along hyphae, compared to 3% of the channels colonized by bacteria alone (Supplementary Fig. 5a).

Protist colonization of the pore spaces (in total 261 encounters, dominated by flagellated and some amoeboid groups) was also increasingly enhanced when fungal hyphae were present in the pore spaces during the first 20 days after inoculation, until the artificial waterlogging event prior to day 28 (presence of hyphae*time $F = 4.5$, $p = 0.03$, Fig. 4b). Protists were almost five times more abundant in the pores containing hyphae, and their colonization depth into the channels was increased 4-fold. After 20 days, 23% of the channels contained protists, in 68% of those together with hyphae (Supplementary Fig. 5b).

Bacterial and protist dispersal increased over time until we performed the waterlogging event before measurement at day 28, equalizing water levels in channels with and without hyphae, and consecutively also their bacterial and protist colonization (Fig. 4b). After the waterlogging event, we let the chips dry out and found, contradicting our expectations, that channels containing fungal hyphae did not retain water better than the ones without hyphae. We also investigated the dynamics of hyphal influence on bacterial dispersal using a third approach, by quantifying the colonization of droplets of condensed water that spontaneously and frequently formed within the pillar system, analysed in one chip (Fig. 4e, Supplementary Movie 15). Initially, most water droplets were sterile, and ca. 25% of them started to form along hyphae. The first water droplets colonized by bacteria were almost exclusively those containing hyphae. Over time, drying and rewetting events caused droplets to unite or split so that bacteria-only droplets started to occur more frequently.

**Microhabitat formation.** In addition to microorganisms, abiotic components also entered the chips: mineral particles, identified with in-situ Raman scattering microspectroscopy as quartz (Supplementary Fig. 2), as well as soil solution. This enabled us to study how microorganisms interact with soil particles and contribute to shaping the dynamic soil pore space. We observed the formation of new microhabitats inside and around newly formed soil aggregates (Fig. 1a–c, h–i, Supplementary Movies 1–2, 7-8), changing the original pore connectivity and pore size distribution.

We recorded particles being dragged along moving water inside the chip in evolving meandering stream patterns, and bacteria being strongly displaced (Supplementary Movie 16). As a result of the water mass flow, stream channels in between the mineral particles developed and were re-shaped over time. Particle movement in the streams was quantified via automated particle tracking (Fig. 5, Supplementary Movies 17, 18; tracked particles and original video, resp.). Particles in the presented video (recorded at a minimum displacement distance of 20 µm per frame) moved with an average speed of 65 µm/s. The average speed of the fastest 10% of these particles was 138 µm/s, most of which moved through one of the two clearly visible water flow paths in Fig. 5a, while the largest part of the recorded area showed little or no water movements.

We also recorded effects of biota on the soil physical structure: fungal hyphae directly changed the pore space shape by squeezing between the polydimethylsiloxane (PDMS) and the glass of the chips with help of hyphal tip forces (Fig. 1d), or by blocking passages in the chips' pore spaces themselves (Fig. 1e, g), and indirectly, by constituting barriers for mineral particles that accumulated along the hyphae during water movement of drying and rewetting events (Fig. 1c, h). Hyphal colonization restricted microbial dispersal of larger organisms such as protists and nematodes by occupying and obstructing the access to free pore spaces (Fig. 1g, Supplementary Movies 6, 19). Both nematodes

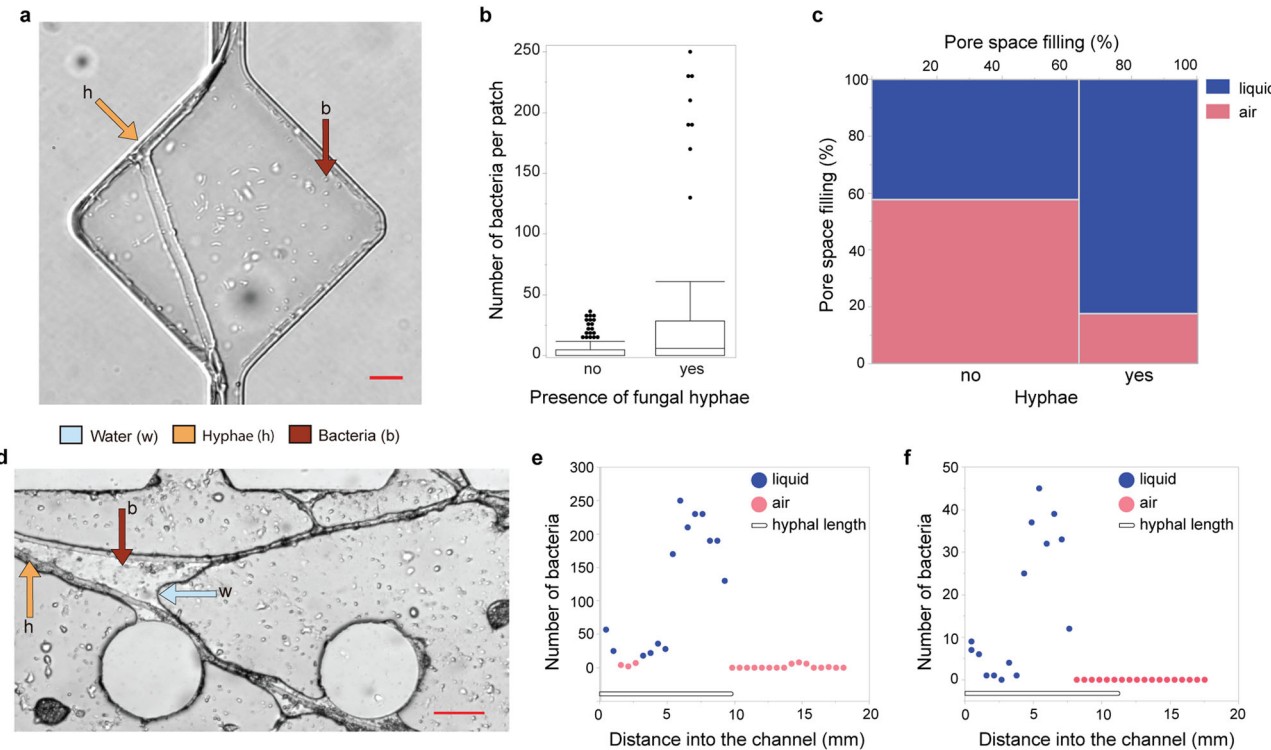

**Fig. 3 The influence of fungal hyphae on the dispersal of bacteria in the field. a** Example of a diamond-shaped widening along the channels which served as the basic entity to calculate bacterial abundance depending on the presence of a hypha. Scale bar = 10 μm. **b** Abundance of bacteria in diamond-shaped openings depending on presence of a fungal hypha, n = 4 channel pairs × 33 openings in a paired ANOVA, boxplot with median, quartiles and outliers. **c** Contingency diagram showing the occurrence of air or water in diamond-shaped openings depending on the presence of a fungal hypha. The frequency of channels containing hyphae is shown along the x-axis the, the frequency of channels containing liquid or water is shown along the y-axis, 264 observations in total. **d** Example of fungal highways developed within the soil chip's pillar system, where hyphae drag water films with them, allowing bacterial transgression. Scale bar = 50 μm. **e, f** Bacterial abundance along a fungal hypha protruding into an initially air-filled channel. Bacteria were quantified per diamond-widening, each dot represents a diamond. Red dots represent diamonds that remained air-filled after 2 months in the soil, blue dots represent soil-solution filled diamonds at examination. The line beneath the data curve represents the extent of the hypha in the chip. Data derives from chips of Expt. 1 that were buried in soil and initially air-filled.

and larger protists were frequently observed having difficulties passing hyphae in the chip with its low height of 7 μm, especially if the hyphae occupied narrow constrictions in the solid pore matrix of the chip. Phagotrophic ciliates displayed an active hunting behaviour, pushed into aggregates and moved them, thus also creating new passages (Supplementary Movie 20). During drying processes we saw distinct developments of water menisci connecting structures (Fig. 1a), especially around fungal hyphae (Figs. 1f and 3d), which increased hydraulic connectivity and maintained the potential for bacterial motility. The cells that died inside the chip laid the base for an initial organic matter build-up, and necromass inside the system was recycled within the food web which was clearly visible when bacteria accumulated around fractured cells and organisms (Supplementary Movies 21 and 11).

## Discussion

Despite the infinite number of pores present in soil, soil microbes usually occupy only a fraction of them[27], and they need to navigate across these in search of food and suitable environmental conditions. Investigating the soil microhabitats at a relevant scale will give deeper insights into soil processes such as carbon cycling. The soil chips, colonized by a rich microbial community, constitute literal windows into the soil, allowing us to monitor soil processes in real time, and multi-level interactions among and between microorganisms and their habitat (Fig. 1). We examined the effect of principal pore space characteristics (Fig. 2) and

showed that fungal hyphae frequently increased pore connectivity and dispersal of water-dwelling organisms (Figs. 3, 4), which is likely of high relevance for their colonization success in an ever-changing pore space system. We demonstrated that the dynamics of changing microhabitats are caused by both physical forces and biological activity (Fig. 5).

The pore space geometry inside the soil chips affected the colonization capacity of fungi, showing that fungi experience difficulties navigating through acute angles and geometries that are not in line with their initial growth direction. This confirms previous results from laboratory experiments[26,28], which we here extend to species from complex natural inocula. The variation found in our results suggests however that fungal reactions to microspaces can depend on factors not specifically tested in this study, such as priority effects by the order in which individual species entered the channels[26], or seasonal variation. The water-dwelling organism groups bacteria and protists were, as expected, not affected in their principle dispersal capabilities by the channel geometry. It has been suggested that fungal hyphae prefer growing in air-filled pore spaces[29], but our study did not confirm this to be generally true. However, we did find that water-dwelling organisms are strongly restricted in their dispersal ability in air-filled pore spaces (Fig. 2). Under these dry soil pore conditions the phenomenon of 'fungal highways' becomes important, where hyphae facilitate the dispersal of motile bacteria along them[30,31].

While fungal highways have previously been demonstrated in laboratory settings[32,33] and indirectly in-situ[34], we have produced

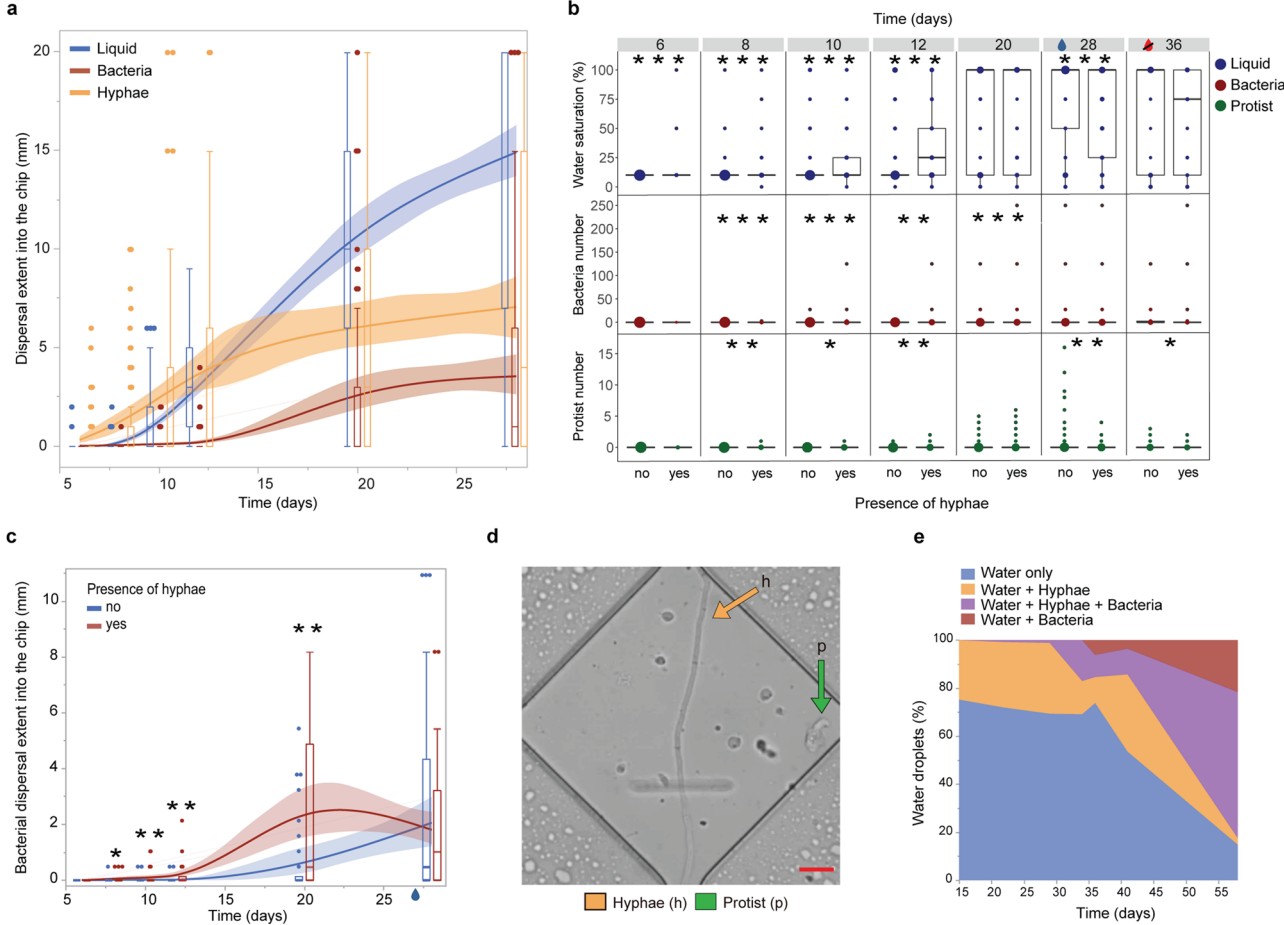

**Fig. 4 Time-resolved investigation of the influence of fungal hyphae on pore wetting and microbial dispersal recorded in Expt. 3: soil incubated on initially air-filled chips in the laboratory. a** Dispersal extent of fungal hyphae, water films (liquid), and bacteria into the 'diamond' channels, mean and 95%-confidence interval combined with boxplots with median, quartiles and outliers, $n = 3$ chips × 36 channels. **b** Presence of water (liquid) and abundance of bacteria and protists in diamond-shape widenings over time, depending on the presence of a fungal hypha. Prior to day 28, we performed a waterlogging treatment to the inoculation soil that saturated most parts of the chips' pore spaces, indicated by the blue droplet in **b** and **c**. Between days 29 and 36, chips were exposed to drying without any additional watering, indicated by the strikethrough red droplet in **b**. Boxplots with median, quartiles and outliers and the dot size indicating the number of observations, $n = 3$ chips × 1188 widenings. **c** Bacterial dispersal extent into channels with diamond-shaped widenings, depending on the presence of fungal hyphae, boxplots combined with a curve of the means and 95%-confidence interval, $n = 3$ chips × 36 channels. Prior to day 28, we performed a waterlogging treatment to the inoculation soil that saturated most parts of the chips' pore spaces, indicated by the blue droplet. **d** Example of a diamond-shaped widening along the channels with a protist along a hypha. Scale bar = 10 μm. **e** Quantification of the presence of hyphae and bacteria contained within spontaneously formed water droplets within the entry system of an initially air-filled chip. Humidity of the inoculation soil was kept equal, no waterlogging or drying was applied. Stars in **b** and **c** denote statistically significant differences at *$p < 0.05$; **$p < 0.01$; ***$p < 0.0001$.

direct microscopic footage of hypha-mediated bacterial dispersal proceeding from natural conditions into a pristine pore space, quantified that footage, and elucidated the sequence of processes and causes over time (Figs. 3 and 4). The presence of fungal hyphae facilitated the dispersal and population growth of water-dwelling soil microbes, including protists, severalfold. Our results suggest that enhanced pore filling with soil solution, rather than newly created elongated liquid films along the hyphae, is the responsible mechanism (Supplementary Movies 12–14), and single remaining air bubbles were observed to be insurmountable obstacles for bacteria (Supplementary Movie 22). The higher levels of wettability in channels containing fungal hyphae could be explained by the fungi producing exudates, in a manner similar to how bacterial extracellular polymeric substances have been shown to be responsible for increased water retention in micropore spaces[35]. Dispersal via fungal hyphae is hypothesized to explain the maintenance of costly flagella for soil-dwelling bacteria, even in soil with generally low water content[36]. In this study, bacteria were found to colonize even channels without hyphae, but in fewer cases, and cell numbers commonly remained lower in those channels (Fig. 4b, c, Supplementary Movie 14).

Bacterial abundance is known to be generally higher in the mycosphere than in bulk soil[33]. The numbers of bacterial cells in the chip pore spaces are a product of their dispersal and their growth inside the pores. Laboratory experiments have shown how mycosphere bacteria are likely to have greater access to nutrients[37–39], both in the form of exudates and nutrients released during fungal degradation of organic matter. The pore spaces of the air-filled chips were initially nutrient void, and at least part of the increased bacterial abundance along hyphae in our experiment may be explained by hyphal-fed population growth inside the chip, rather than solely by means of dispersal via the hyphae.

The potential importance of fungal hyphae for dispersal of other soil organisms, such as protists, has previously been hardly recognized, and our results aim to bring attention to this

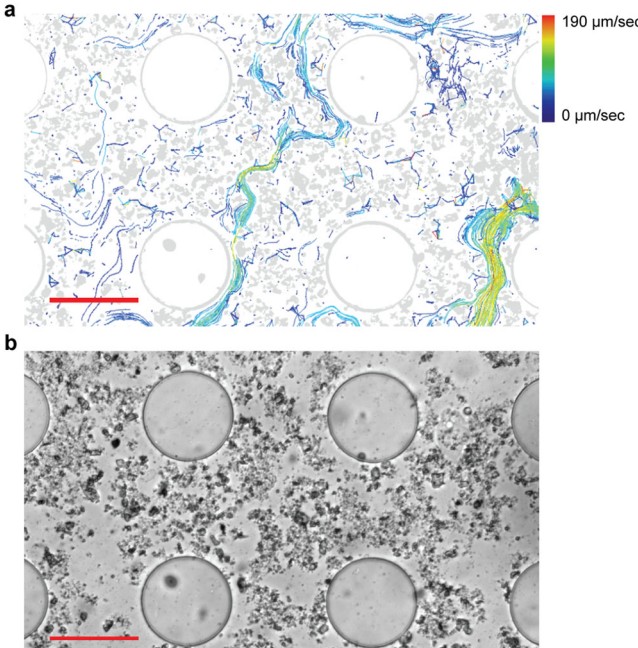

**Fig. 5 Preferred water flow paths in the soil matrix within the chip, revealed by tracking of displaced particles. a** Threshold image and tracked particle visualization of a 45 s long real-time movie sequence (ImageJ plug-in TrackMate version 3.8.0.; Supplementary Movie 17). Colors of the lines following the tracked paths indicate the mean velocity of the particles. The large round structures are pillars of the chip, smaller particles are soil minerals that were dragged into the chip. The recorded water flow was caused by drying of the chip through the adjacent soil layer. **b** Original bright-field image (Supplementary Movie 18), recorded from an initially air-filled chip buried in soil of Expt. 1. Scale bars = 100 µm (**a**, **b**).

unexplored field of research. The larger eukaryotic cells would not be able to travel in water films along hyphae, but they can take advantage of the increased water filling in larger pores that hyphae generate. The influence of fungal hyphae on general microbial dispersal dynamics is thus greater than previously acknowledged. Especially important, our data can demonstrate that this phenomenon is not a rare, anecdotal event but rather, under intermediate humidity conditions in the pores, occurred in up to 32% (bacteria) resp. 16% (protists) of 108 examined channels. In a real soil pore system, this priority dispersal should exhibit a large competitive advantage, as freshly opened soil spaces (e.g., via bioturbation[40]) likely contain free nutrients and degradable necromass.

Soil-inoculated chips hold an especially high potential for future studies involving protists, such as their involvement in trophic networks. Although it is well known that many protists feed on bacteria and control microbial populations[41], the details of dietary requirements of most soil protists, and their functions in soil processes, have been difficult to study directly[42]. Our soil chips create new possibilities for visual and time-resolved monitoring of food-web structures and may in the future identify food-web members via single-cell genomics on extracted material, increasing our understanding of protists' involvement in soil nutrient cycles.

Both macro- and microaggregates are in permanent turnover[43], and much of the soil pore space is permanently being reshaped. The largest effects on microhabitat modification are usually caused by physical forces. Preferential flow paths are the main channels for water movement in soil, and their chemical composition can be very different from the soil matrix located in smaller pores, because only a fraction of water and solutes passes

from the preferential flow path to the surrounding matrix[44–46]. Our results illustrate that microbes will experience very local levels of habitat stability and nutrient supply. The dynamic habitat landscape might promote organisms capable of anchorage or biofilm formation. Our results constitute unique records of water movements in and around mobile and realistic soil microaggregates[1] which result in the development of distinctly different characteristics in microhabitats located only a few micrometers away from one another. Other current studies are commonly performed at spatial scales larger than soil micropores, since most of the methods used (e.g., magnetic resonance imaging, neutron computed tomography, and X-ray tomography) produce insufficient image resolution[47–49], and are often incompatible with *live* imaging of undisturbed biota activity during measurements which leaves crucial knowledge gaps regarding flows and solute transport in soils at the low micrometer and nanometer scale where microbial activity is confined.

Visualizing how flows influence the microbial environment can help us predict the way microbes react to and interact with different flow movements in the soil. Chip studies on time-resolved microbe-habitat interactions constitute a valuable addition to studies in 2-D thin sections of soil microbe distribution[15], e.g., for investigations of bacterial biofilm settlement in relation to different shear forces in a simulated sand pore space[50]. Aufrecht et al. successfully identified the spatial distribution of fluorescent bacterial strains when inhabiting a microfluidic chip with water flows around a fixed solid PDMS matrix, while we here use a pore space combining fixed and mobile solids (the PDMS structures and the soil particles, resp.) which allow us to record spatial changes in the pore space as a response of water flows. However, it must be considered that the measurements and observations of the present study correspond to flow properties around microaggregates formed inside a synthetic porous medium with surface roughness and charges that might differ from those found in natural soils. This could be improved by harboring more soil material and having a longer stabilization phase, allowing organic coatings to form on the chip surfaces. Despite the simplifications of this experiment, we believe that the knowledge obtained will be of great significance for an increased understanding of water flows, such as during drying and rewetting inside soil aggregates, and their impact on soil biota and their functions.

Bioturbation of soil is commonly ascribed to larger soil fauna[51], while insufficient data is available on the role of microorganisms[52]. The strong influence of fungal hyphae on both opening and blocking of passages in soil makes them important 'ecosystem engineers' for microbial microhabitats. Hyphae are able to exert considerable pressure at their tips[53], and when many hyphal tips join they can vigorously open passages in their surrounding (Fig. 1d)[26]. The frequency of hyphae blocking pore space passages for both soil components and organisms recorded in the soil chips is a likely overestimation when aiming to compare to their occurrence in real 3-D pore systems, because of the chips' uniform and low height (7 µm), but our results illustrate the principle that hyphae can influence pore connectivity at necks and constrictions. We found, unexpectedly, not only hyphae but even protists forcing new passages through aggregated soil in their hunt for food (Supplementary Movie 19). This demonstrates the importance of the interplay of the microbial community with their fluctuating microenvironment, at different scales, for important large-scale processes such as nutrient cycling.

We see microfluidic chips as a promising tool to deepen insights into soil ecology by observing biological processes in a transparent and geometrically realistic environment, making it possible to track large numbers of individual cells in real time, including traditionally unculturable species[23], and as an optimal platform for connecting research frontiers in the fields of soil

physics and soil ecology at relevant scale. Nevertheless, soil ecologists should remain aware of the limitations of these highly artificial systems, where current technology only provides a pseudo-3-dimensional space, and the fabrication material lacks many properties of soil minerals: the chips are unnatural in terms of their chemical composition (silicone-based polymer) and surface properties (smooth, inert, and either homogenously hydrophilic when wet, or homogenously hydrophobic when dry). Nevertheless, the chips represent an improvement in realism in terms of their physio-spatial properties because they fragment the microbial habitat, their minuscule channels generate a large surface structure with strong surface tension, capillary forces, adhesion, and viscous drag; and liquids moving through the structures exhibit a laminar flow, just like real soil solution in small pores[21].

Compared to the more frequently used lab-strain inoculation of microfluidic chips, soil inocula contain many organisms that interact with each other, as well as solutes and mineral particles diffusing into the chips. This can thus add several layers of complexity into controlled laboratory experimental systems and may become an important link between field and laboratory experiments in microbial ecology. We present two different ways of using soil as ecosystem inoculum, either in-situ incubation in a natural soil ecosystem, or else taking soil inoculum into the laboratory. Combining these two approaches may yield the most comprehensive results for many questions: the most realistic picture may be drawn from in-situ inoculation, where chips are exposed to natural oscillations in humidity, temperature, plant photosynthate inputs, and changes in the microbial species pool. Fungi can interconnect the natural soil with the chips' interior and move nutrients into it, especially in mycorrhizal symbiosis with surrounding plants. On the other hand, experiments using soil inoculum in the laboratory are easier to accomplish, control, and manipulate, and may be monitored with time resolution. To combine advantages of both, microcosms could be constructed with larger amounts of soil, including vegetation, placed around the chip in a large Petri dish.

Micro-engineered chips could in the future be successfully combined with other techniques (e.g., microspectroscopy, epifluorescence microscopy, single-cell sequencing) to incorporate high-resolution information about the chemical and genetic characteristics of the different soil components[54]. Future studies may investigate directly and at microscale how abiotic components (e.g., pH or toxic compounds) affect different microbial groups; how the diversity of microbial communities is linked to different soil ecosystem functions such as soil carbon sequestration; and the ecophysiology of important but understudied organism groups such as soil fungi and protists. We even see a high potential in microfluidic chips to bring soils closer to society, as visual experience may increase interest and concern for an ecosystem in need of protection.

## Material and methods

### Chip design
We used our micro-engineered silicone chip termed the "Obstacle chip"[26], representing a proxy of a soil pore space system containing different sets of microstructures. The chip consists of an artificial pore system open on one side for inoculum, and it is designed to investigate the growth and dispersal behaviour of soil microbes (Supplementary Fig. 1a, b). The chip's pore-space dimensions are optimized to match the dimensions of fungal hyphae, with structure widths ranging between 4 and 100 μm, and a uniform height of ~7 μm to aid microscopy, since cells are located in the same focal plane and rarely overlay. It contains five different geometric sections accessible by soil microbes via a common entry area

(Supplementary Fig. 1a). The entry area consists of an open area with round pillars of 100 μm diameter at a separation of 100 μm, holding up the chip's ceiling. It was cut open longitudinally with a scalpel prior to bonding (see below, section 'Chip fabrication'), ensuring direct contact of the soil with the chip's interior. The inner section comprises a combination of differently shaped channels and obstacles constituting five experimental sections, of which two were systematically examined in this study: (1) Section C: A set of channels with sharp corners of three different types ($n = 12$, randomly distributed): zigzag channels (90° turns with all channel sections at ±45° angle from the main growth direction), meandering square channels (90° turns with each section oriented in either the main growth direction or perpendicular to it), 'z'-shaped channels (sharp corners diverting 135° from the previous growth direction, with channel sections in the main growth direction and at angles of 45° and 135° from it); (2) Section D: So-called 'diamond channels,' with a repeated combination of 10 μm wide and 400 μm long straight channels alternating with 140 μm wide diamond-shaped widenings. This channel type is replicated in 36 channels, each containing 33 diamond widenings. The widenings were used as quantification units to count bacteria and protist cell numbers, and for determination of liquid ingression, for the experiments on dispersal via fungal hyphae. Section A of the chip contains systems of hexagonal pillars of different diameters, Section B consists of straight channels with different widths, and Section E contains two types of obstacle courses comprised of complex structures. Those and the entrance system provided space for general observations. The design was constructed in AutoCad 2015 (Autodesk), in which patterns within experimental Sections A and C were randomized using a custom script from UrbanLISP (http://www.urbanlisp.com).

### Chip fabrication
The microfluidic chips were moulded in PDMS on a photoresist master defined by UV lithography and bonded to glass slides, according to Aleklett et al.[26]. The master was made by spin coating a thick negative photoresist (SU-8 5, MicroChem Corp, USA) on a glass plate for 60 s at 1250 rpm. This generated a photoresist layer of ~7 μm. The photoresist was soft baked for 5 min at 90 °C on a hot plate, patterned by UV exposure (Karl-Suss MA4 mask aligner) and post-exposure baked. It was then developed for 3 min in mr-Dev 600 (MicroChem) and finally rinsed with isopropanol (VWR International). The PDMS slabs were produced by thoroughly mixing a PDMS base and a curing agent (both Sylgard 184, Dow Corning, USA) in a 10:1 ratio, followed by pouring the mix onto the master in a 4-mm-thick layer, and degassing it in a vacuum chamber at −25 kPa for 45 min. Then the PDMS was cured in an oven for 2.5 h at 60 °C. Once cooled, the PDMS was cut slightly larger than the designed pattern, covering an area of about 40 × 65 mm, and cut though the entry system, creating a lateral opening to the chip along the pillar system.

The PDMS slabs were bonded to glass slides. Glass slides, 55 × 75 mm and 1 mm thick (Thermo Scientific), were first cleaned with acetone, 75% ethanol and deionized water, and then dried under an air-blower. The pieces of PDMS and the glass slides were treated separately in an oxygen plasma chamber (Diener Electronic Zepto). For each chip, a glass slide was exposed to oxygen plasma under UV light for 1 min, followed by exposure of the PDMS piece for 10 s. Once both samples were plasma-treated, they were immediately brought in contact with their activated surfaces facing each other, and gently pressed to each other in the centre parts of the chip. To avoid collapse of the ceiling of the entrance, none of the chip edges were pressed. The chips were heated on a heating plate for ~15 s at 100 °C to ensure a proper

bonding. After another 15 s, the chips with liquid treatments were filled with the different media using a micropipette, taking advantage of the PDMS's temporary hydrophilia following plasma treatment so that liquids were readily drawn into its structures. The chips were filled with one of the following three treatments: (1) deionized water, (2) liquid malt medium, a complex medium to provide a nutrient-rich environment including reduced sugars such as disaccharide maltose and in lower proportion nitrogenous components such as peptides, amino acids purines and vitamins (malt extract for microbiology, Merck KGaA), or (3) chips were left empty, i.e., air-filled. The eight chips filled with liquid were then placed in a vacuum chamber for 30 min at −25 kPa to remove any bubbles. Finally, the chips were kept in sterile Petri dishes, sealed with Parafilm and stored overnight in a cold room before being dug down into or inoculated with soil.

**Expt. 1: in situ incubation of chips**. To evaluate the effect of different nutritional conditions on colonization of the soil chips by microbes, we evaluated three pore space filling treatments: (1) deionized water, (2) malt extract medium, or (3) air; $n = 3$ chips per treatment. The experimental site was a small grove of deciduous trees in the city of Lund, Sweden (55° 42′ 49.5″ N, 13° 12′ 32.5″ E; Supplementary Fig. 1c). The season chosen for burial of the chips was early autumn (October 2017) to guarantee a moist soil during the experiment. Groups of replicates of all three chip treatments were buried randomly within the inner parts of the grove ($n = 3$ chips per filling treatment). The litter layer was removed, and $20 \times 20$ cm holes were carefully dug into the ground with a spade. The chips were placed horizontally in the soil at a depth of 10 cm in which the PDMS chip was facing up and the glass slide down. Horizontal placement was chosen to probe a single stratum of the soil, serving as a comparable inoculum to the whole of the entry system, and to aid nondestructive recovery. The soil was carefully placed back in its original orientation, and the litter layer was placed back. A string attached to each chip was placed with its opposite end above the soil surface and attached to a pin, to guide future retrieval. There was a minimum distance of one meter between each chip replicate.

Preliminary experiments had shown that a 2-month incubation period would grant the colonization of different types of soil microorganisms and minerals, and a stabilization of the inner environmental conditions between the soil chip and the surrounding soil. Thus, after 64 days (December 2017), the chips were collected by carefully removing soil around the string leading to each chip. We carefully kept the adjacent soil atop the glass slide along the opening of the chip, to keep our artificial pore system connected to the real soil pore system, and to avoid such disturbances as hyphal tearing or evaporation of the liquid inside the chips (Supplementary Fig. 1d). We cleaned the chip windows by softly wiping them with a clean wipe and deionized water. Samples were carefully transported to the microscopy facilities, located adjacent to the burial site. The chips were harvested one at a time and analysed under the microscope immediately after collection and cleaning.

We recorded the presence or absence of the main soil microbial groups in the entry systems and in the different channels, including their furthest extent into the chips, with help of the internal rulers.

To analyse the effect of fungal hyphae on bacterial abundance, we recorded real-time videos slowly scanning along the whole length of the diamond-shaped opening channels (each 33 diamonds, Section D in Supplementary Fig. 1a; Fig. 3). The rather sparse hyphal colonization allowed us to select pairs of channels where in the first channel a hypha had proliferated far

into the channel, combined with a directly adjacent channel without hyphae, $n = 4$. In each diamond-shaped widening we counted the number of bacterial cells, the presence or absence of fungal hyphae, and the presence or absence of liquid. After completion of all measurements, the chips were left uncovered at room temperature for 60 min to initiate air drying in the adjacent soil, in order to observe the real-time effects of drying on organisms and particles in the pore space system of the chips. The adjacent soil was re-wetted by adding 400 μl of water. The water inside the chips corresponded to the adjacent soil pore water, regressed upon evaporation, and refilled the chip structures upon rewetting of the adjacent soil.

**Expt. 2–3: laboratory incubation of soil on chips**. In a complementary approach, we collected soil from a lawn in Lund, Sweden, at 10 cm depth, and placed 5 g of this soil in front of the entry system of the chip. Chips received the three nutrient condition treatments as described above, air, water or malt medium ($n = 2$, Expt. 2). An additional set of air-filled chips was studied to quantify fungal highways ($n = 3$, Expt. 3). Chips were monitored under the microscope after inoculation, observation was documented with images and videos. Chips were kept in sealed Petri dishes with wet cotton cloths to maintain high humidity and were taken out for analysis only. The soil inoculum on the chips and the interior of the chips were kept moist with 500 μl of water added to the soil once a week. The artificial waterlogging event in the chips of Expt. 3 ('fungal highways') was achieved by adding a total of 2 ml of water to the inoculum soil over the course of a week, and the drying event was achieved by discontinuing the watering.

During Expt. 2, we recorded the abundance and the furthest extent of bacteria, protists (including the morpho-groups ciliates, flagellated, and amoeboids), and the extent of hyphal colonization into the diamond section over time. After 2 months of incubation, we measured the furthest extent of colonization into the angled channels for the organism groups bacteria, fungi, and protists. During Expt. 3, we recorded the presence and the furthest extent of hyphae, liquid, bacteria, and protists in the diamond channels over time. We also recorded the number of protists, bacteria (in categories 0, <5, <50, <250, <500), and the percentage of the pore filled with liquid (in categories 0, <10, <25, <50, <75, 100) in each diamond-shaped pore.

**Microscopy and image analysis of the chips**. All visual inspection and imaging were done with an inverted microscope (Nikon Diaphot 300) with 40–400× magnification. Bright-field images and real-time videos were recorded through the microscope with a digital camera (USB29 UXG M). Presence and abundance of organisms and liquid was counted by eye. Active bacterial cells were counted when exhibiting nonrandom movement, allowing to record cells down to a size of circa 0.4 μm in movement. The real cell number may be underestimated since small cells may have remained undetected at rest. The absence/presence of the main soil microbiota groups (fungi, bacteria, protists, nematodes, and microarthropods) was recorded in the entry system and in the different channels. Where present, the maximum individual extent of each soil biota group into the chip was recorded in the turning and diamond channels (Section C and D). Videos and images of particles and microbial interactions were recorded in the entry system and in the channels and pillar systems of the chip. All images shown in Fig. 1, except for Fig. 1c, h, were extracted from videos (Supplementary Movies 1–6, 8) using Adobe Premiere Pro CS6. All images shown in Figs. 1 and 3 were cropped in Adobe Photoshop CC (v. 20.0) and the image from Fig. 4 was cropped in Adobe Photoshop 2020 (v.21.2.4); contrast and brightness were adjusted to obtain optimal visual data. Scale bars were added with

ImageJ Software (v. 1.56 h, NIH). False-color image legends were produced in Adobe Photoshop CC (v. 20.0). The central illustration and the final assembly of other elements of Fig. 1 were created in Adobe Illustrator 2020. Images shown in Fig. 5 were captured using the ImageJ plug-in TrackMate version 3.8.0.

**Velocity analysis of particles in water flow paths**. The velocity analysis of soil particles in soil solution streams was performed in ImageJ version 1.52i. A 48-s video clip of a drying event inside the chip was converted to an 8-bit file and subjected to a threshold of gray values from 0 to 175. This range was selected because it showed to be optimal in the number of moving particles captured. Particles were tracked with the Image J plug-in TrackMate version 3.8.0. A LoG detector, which applies a Laplacian of Gaussian filter to the image, was used to detect the particles. The parameter 'Estimated blob diameter,' which should have approximately the same size as the analysed particles, was set to 2.76 μm. The radius of the particles considered by the software then ranged from one-tenth to two times the radius. This value represents the quality of the detection and must be a positive number with higher numbers representing higher quality. This size range was chosen because particles smaller than 2 μm were frequently lost in the tracking algorithm, and particles larger than 6 μm commonly were aggregates that could break apart and move in parts. The quality histogram threshold was set above 37. The particles were tracked using the simple Linear Assignment Problem (LAP) tracker function, with linking and gap-closing conditions of 30 and 40 pixels respectively, and a maximum frame gap of 0 frames (frame rate: 18.3 frames per second). Velocity tracking in Fig. 5 is visualized as color-coded mean velocity tracking lines, showing individual velocities of the particles.

**Raman scattering microspectroscopy in the chips**. Raman scattering spectra of particles in the chips were recorded using a long working distance ×50/0.5 objective of a LabRAM HR Evolution confocal Raman microscope, equipped with 785 nm diode excitation laser, 600 g/mm diffraction grating, and front-illuminated thermoelectrically deep-cooled CCD camera Sincerity (Horiba Scientific). The confocal mode of the microscope was used, employing 50 μm diameter pinhole. Spectra in the spectral region 200–970 cm$^{-1}$ were acquired with an acquisition time of 30 s and 32 accumulations, at full excitation laser power (neutral density filter $T = 100$ %, $P_{785} = 100$ mW). Spectra of the particles in the chip were compared with mineral reference spectra and identified as quartz. Other peaks in the spectra are assigned to the PDMS (Supplementary Fig. 2).

**Statistics and reproducibility**. Data presented in Figs. 2–5 (Supplementary Data 1) was statistically evaluated in JMP Pro 15.0 (SAS Institute Inc., Cary, USA) with the significance level set to $\alpha = 0.05$, two-sided tests. Analysis was performed on log-transformed or square-root data if necessary to meet normal distribution of the residuals.

*Expt. 1*. The influence of the chip-filling treatment and the different channel shapes on the maximum colonization distances of fungal hyphae, bacteria, and protists was evaluated via a restricted maximum likelihood ANOVA model, with the full-factorial combination of the two factors 'channel type' (three levels: zigzag/square/z-shaped; maximum extent per channel type) and 'nutrient treatment' (three levels: air/water/malt medium), which were full-factorially distributed at each location ($n = 3$ per channel and filling type). The influence of the presence of fungal hyphae on the presence of bacteria in Expt. 1 was investigated in a paired set of channels with a hypha, against directly adjacent channels without hypha, $n = 4$. Data of four channel pairs from one air-filled chip were evaluated by a paired two-way $t$-test. In an alternative approach, individual pores were used as a replication base, $n = 33$ in 4 pairs in a restricted maximum likelihood ANOVA model depending on the presence/absence of a hypha and with pairs as random attribute factor, which produced similar results. The influence of the presence of fungal hyphae on the presence of water in the diamond widenings of the initially air-filled chips was analysed with a $2 \times 2$ contingency analysis and evaluated for dependency with Pearson's Chi-squared test.

*Expt. 2*. A restricted maximum likelihood ANOVA model with the full-factorial combination of the two factors 'channel type' (three levels: zigzag/square/z-shaped; 12 of each channel type per chip) and 'nutrient treatment' (three levels: air/water/malt medium), and the factor 'chip replicate' ($n = 2$) as a random attribute, was used to test the influence of nutrient condition and channel shape on fungal, bacterial, and protist dispersal.

*Expt. 3*. A restricted maximum likelihood ANOVA model, with a full-factorial combination of the factors 'time' and 'presence of hyphae', and with the factor 'chip replicate' ($n = 3$) as a random attribute, was used to test the influence of fungal hyphae on water ingression, bacterial, and protist dispersal over time both on pore space ($n = 1188 \times 3$ chips) and at channel replication ($n = 36 \times 3$ chips). As the pore space dataset was strongly skewed due to the initially rare events of microbial colonization, we analysed the data in parallel with non-parametric Wilcoxon/Kruskal Wallis tests and with a logistic regression using the presence/absence of hyphae and bacteria or protists, which gave similar results in terms of which comparisons were significantly different.

**Reporting summary**. Further information on research design is available in the Nature Research Reporting Summary linked to this article.

## Data availability

The datasets generated and analyzed during the current study presented in Figs. 2–5 are available under Supplementary Data 1, and image and video documentations are available from the corresponding author upon reasonable request.

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

## Acknowledgements

We thank Martin Bengtsson for expertise and assistance during the design and production of the chips. E.H. acknowledges funding from the Foundation of Strategic Research (Future research leader grant SSF FFL18-0089), E.H. and P.O. acknowledge funding from the Swedish research council (VR-621-2014-5912), P.O. from the Sten K Johnsson foundation (20200363), E.H. and K.A. from the Crafoord foundation, the Wallenberg foundation and the strategic research environment for Biodiversity and Ecosystem Services in a Changing Climate (BECC).

## Author contributions

E.H. and P.M. developed of the idea and concept of the study. K.A. designed and fabricated the soil chip master, P.O. provided technical assistance and expertise for the design and fabrication process. P.M. fabricated chips, recorded and analysed microbial growth inside the chips. E.H. statistically analysed the data from the chips. P.M. and E.H. prepared figures and movies. C.A. analysed particle velocities. MP performed spectroscopic analysis of minerals in the chips. P.M. and E.H. wrote the first version of the manuscript. P.M., C.A., K.A., M.P., P.O., and E.H. contributed to interpretations of the data, the revisions and the finalization of the manuscript.

## Funding

## Competing interests

The authors declare no competing interests.
