## [Peer Review File · Communications Biology]

Reviewers' comments:

Reviewer #1 (Remarks to the Author):

Recommendation: Major Revision

Overall impression:

This was an extremely interesting and fun paper to read. This work offers a unique perspective on the use of a 'Soil on Chip' platform and demonstrates how these types of platforms can be used to generate new hypotheses about multi-kingdom interactions. I applaud the authors for their bravery in attempting to bury these chips in a natural environment and then 'see' what happens. The authors clearly and honestly describe the limitations of the synthetic, 2D material platform, but make appropriate use of the different shapes in their model system to look the effects of confinement and connectivity on co-colonization and wetting.

I enjoyed reading the paper and found that it stimulated new ideas for my own work about how to test specific hypotheses that were posed, however it seemed to fall short in truly enabling or executing a specific set of experiments that truly prove or disprove these hypotheses or delve into the mechanisms that drive the phenomena that were observed. As a collection of interesting observations that were made, enabled by this platform, I see value in publishing this work, though it is honestly difficult to gauge the value of the contribution. The concept and approach are notable, burying a discovery platform with lots of variables built into it underground and seeing what happens is brave and unique. However, the majority of the manuscript seems to be speculative and observational. Some quantitative analysis is done, most notably the difference in bacterial colonization and hyphal presence, but additional experimentation is warranted to give this manuscript true scientific rigor. Given these observations, it would be worthwhile to replicate that phenomena in the lab using a known fungal species and small collection of soil isolates within a microfluidic chip that contains the diamond design. The dispersal experiment could and should be replicated in the lab in a similar manner. Are the results species dependent? Do subtle variations in the geometry (channel width for the zig zag, etc. impact dispersal)?

Overall, I think that the approach of using microfluidics, buried in the environment, and then bringing them out to look at what happened is a novel notion for generating new hypotheses, but given the emergence of other soil chip platforms, and the missed opportunity to follow up on some of these observations with more targeted and controlled experiments, this manuscript falls a bit short with respect to the novelty and rigor necessary for publication at this time. I recommend performing a few follow up experiments to support the hypotheses generated using the platform and resubmitting. Additionally, addressing the more specific comments below is warranted.

More detailed comments:

1. The language/grammar needs some additional editing to improve the clarity and precision of the writing. I know that this is a bit picky, but some thorough editing and proofreading are needed.

I've picked 3 examples below, but there were quite a few.

a. Line 22, use of the word 'got' with 'was'

b. Line 25, use of the phrase 'corresponded well..., allowing us to eavesdrop live and in a more realistic way...' is a bit of an exaggeration. How did it correspond? Were sizes, gaps, similar to soil? What were you 'eavesdropping' on? Did you measure any chemical species that drove the observed chemical reactions? The observations were made in the lab after the chips were colonized, this isn't truly done in the soil environment.

c. Line 59, 'organization at microscale' should read 'organization at the microscale'

2. Lines 185-211 describe the use of the platform for observing preferential flow paths and potential for understanding solute transport and predicting 'the way microbes react to and interact with different flow movements in the soil.' -- recent work by Aufrecht et al. describes the development of a soil-on-a-chip platform that replicates the porosity and packing of soils and demonstrates the influence of pore scale hydrodynamics on bacterial attachment and alteration of flow. The discussion here should be modified to reflect this work and should be compared/contrasted appropriately.

Aufrecht, J. A., Fowlkes, J. D., Bible, A. N., Morrell-Falvey, J., Doktycz, M. J. & Retterer, S. T. Pore-scale hydrodynamics influence the spatial evolution of bacterial biofilms in a microfluidic porous network (2019). PloS one 14, e0218316)

3. In Figure 2, panels e,f. There is no reason to fit a curve to those plots. Remove it.
4. I did not understand the contingency diagram used in figure 2e. Perhaps there is a more straightforward way of expressing your point.
5. In general, I was confused about what data came from which chip, what were included in the replicates and statistics for the different figures. It would be helpful if some of this information were pulled out of the supplement and added, in brief, to the figures.

Review Report: A window to the underground – micro-chip gives visual access to in-situ soil ecology

Mafla-Endara et al. describe a micro-engineered system that allows researchers to gain visual access to soil ecology *in situ*. Specifically, the authors placed micro-engineered chips, fabricated from the elastomeric polymer poly(dimethylsiloxane) (PDMS), into soil and left them there for a period of two months. They then dug out the chips and made observations regarding what had colonized the architecture, identifying a variety of soil organisms and processes taking place in the microchannels.

Although very simple in nature, this is – to the best of my knowledge - the first account that details an experimental setup whereby one has put soil onto a micro device, which can then be utilized to investigate soil ecological processes *in situ*. It is especially interesting that fungal highways have been observed *in situ* using real soils. This study highlights how lab-on-a-chip technology can help to elucidate (and begin to quantify) processes taking place in soil environments. It is refreshing to see that the authors also appreciate the limitations of the device setup, whilst at the same time informing of the potential future advantages.

Major comments

1. The introduction needs to include an appreciation of how microchips have been used previously to study soil organisms using Lab-on-a-Chip technology, i.e. to put the study into context. This is lacking at the moment. The concept of Soil-on-a-Chip technology, i.e. the use of microfluidic technology to create soil-like environments to understand the complexity of dynamic below-ground biological processes at the microscale, with high spatio- and temporal resolution, is not new and should be discussed in more detail, including the appropriate references, e.g. Stanley et al. Lab Chip, 2016 (“Soil-on-a-Chip”) etc., as well as a reference to Nicolas et al. Appl Environ. Microbiol, 2010 (iChip, first chip to be put into soil; the authors reference this but it should be included in the introduction) etc.
2. The exact chip design is not clear: Figure S1 a and b are very difficult to read and do not detail the exact features of the device (the photograph in b indicates there are many finer structures). In terms of providing scientific transparency and allowing experiments to be reproduced, this needs to be specified in more detail. A clear “to-scale” 2D drawing (including the exact microchannel structure), or alternatively the original CAD file, needs to be included. Also, the specific regions of the device design (A-E) should be enlarged so it is easy for the reader to understand the geometries used in microchannel design etc. Then important feature dimensions, scale bars etc. can be included to complement the description in the text. The central panel in Figure 1 goes some way to addressing these issues, but it is more artistic in nature and not an accurate representation of the exact chip design.
3. The devices are implanted into the soil and then removed ca. 2 months later. How do you ensure that the environment is not altered upon (i) removal from the soil and (ii) transport to the lab? E.g. hyphae are very fragile and could easily be disturbed if there is movement of the soil adjacent to the device opening during transport.

4. How did the authors justify the device design? Is there a logical / mathematical rationale behind the design, or is it just random? It is difficult to really understand / appreciate this as the full device architecture is not provided (see comment 2).

5. Figure 1 (f) refers to the fact that mycophagy is detected in this device. How can the authors prove this is mycophagy? There looks to be some debris surrounding the hypha in video 5, but can one say absolutely that this is debris that originated from the hypha and that bacteria are feeding on it? Also, the authors claim that a nematode is trapped by hyphae in Figure 1 (g). How can the authors prove that the nematode is indeed trapped? The video does not show that the nematode is unable to exit the box, it is sat immobile throughout the duration of the video (video 6). Equally plausible is that the hyphae were already there when the nematode entered. PDMS is an elastomeric polymer (i.e. flexible), and it would be relatively easy for the nematode to squeeze past hyphae. Interestingly, however, it seems that the nematode is feeding on the hypha in this video, which could be a point to include somewhere in the manuscript.

6. Please include for all figures what “n” and the error bars represent.

7. Scale bars need to be added to each of the videos. I also find it unusual to include “©SoilChip” on each of the videos and feel this should be removed for a scientific manuscript.

Minor comments

Line 41: “Soil” not “soils”

Line 42: Reconsider English

Line 47: should read “extraordinarily species-rich”

Line 48: “...many hungry microorganisms.” Please use more scientific terminology

Line 76: at the micrometer scale

Line 92: add a reference to Held et al. Fungal Biol, 2011 and/or Stanley et al. Integr Biol, 2014 (first studies looking at fungal growth and interactions with bacteria in microchannels environments, which discuss the advantage of compartmentalization)

Line 93: can the authors justify why they use malt medium over other media such as potato dextrose broth?

Line 97-98: “10 μm wide channels of different angles” – what does this mean exactly?

Line 96-100: English needs rewording as it is not clear at present. Also, the authors should refer to a supplementary figure (see my major comment #2 above) in order to help the reader understand what they mean to convey exactly. It is a little confusing at present.

Line 101: Why horizontally and not vertically, for example? Please provide a scientific justification for this.

Line 102: Why 64 days? Again, please provide a scientific justification for this time frame.

Line 103: “promptly examined under the microscope”. What does this mean? Same day? Were the samples stored or the temperature regulated to represent the external conditions? Perhaps the comparatively warm temperature of the lab cf. the below-ground soil floor could influence the observed interactions?

Line 128-129: The authors mention that there are hypha-free channels directly adjacent to channels containing hyphae. How do they ensure that there are hypha-free channels, which can be directly compared to the hypha-occupying channels? This is unclear, as it

is not possible to visualize in which channels these measurements were conducted. Is it just random? Or perhaps the hypha-free channels induce a bias, i.e. exclusion of bacteria due to channel architecture rather than absence of a hypha. This is not clear and needs to be addressed.

Line 145-146: English needs revising

Line 148: typo

Line 158: Do the authors give a reason why fungal hyphae avoided chips with malt agar cf. chips filled with water/air? Do the authors know what fungal species are growing in their chips (perhaps this would give some insight as to why)? This is quite surprising, as other published work show that fungal hyphae grow readily in nutrient-filled microchannels (e.g. Ghanem et al. ES&T, 2019 and Stanley et al. Integr Biol, 2014)

Line 161: Please show the data, at least some representative microscopic overviews if possible. This is very interesting and any scientific statements, such as this, should be backed up with data.

Line 400-401: I do not understand what the authors mean with the terminology “water retention bridges”, please explain.

Supplementary Information: Methods

Line 4: photograph

Line 46: specify glass thickness and supplier information

Line 68: What is the malt medium exactly? Please specify.

Reviewer #3 (Remarks to the Author):

Review of "A window to the underground: microchip gives visual access to in-situ soil ecology"
Mafla-Endara et al.

In this paper, the authors present results related to the microbial colonization of a micro-chip which represent a model for empty spaces in soil, designed to encompass different complex structures mimicking potential features of soil. After 2 months of exposure to the natural environment, the micro-chips were recovered and observed mainly with optical microscopy to observe how different microbial groups had invaded its structure. A wide variety of microbes was observed in the chip, together with inert soil particles which also invaded the structure. Together microbes and particles created new microenvironments. The paper also provides some results on the dispersal of bacteria along fungal hyphae, the preference on dry/wet substrate for different microbial groups, and observations on preferential path flows inside the colonized chip.

I enjoyed reading this paper and it is relevant for the field of microbial ecology in soil. While PDMS chips have been used widely in other biological fields, the particular chip used here and its in-situ deployment in soil for observing natural assemblages is novel. I find the multiple observations of natural communities inside the chip and the videos provided as supplementary very exciting and inspirational for future research, as they raise many questions on microbial interactions in soil. As such, this paper would surely stimulate further research into the topic. However, the weaknesses of the paper are in my opinion that it provides more of a collection of observations without directly answering a strong scientific question. The results related to bacterial dispersal along hyphae and preference of microbes for dry/wet substrate are sound but have already been demonstrated in other contexts, as mentioned by the authors themselves. The results on preferential path flow in structured environment seem quite preliminary, with only one example of flow visualization using particle tracking, and no real quantitative estimation of the impact of the measured speed on the transport of a diffusive nutrient. This flow visualization thus looks like a demonstration of feasibility to me.

I now move on to more specific comments on the paper which aim at improving clarity, readability and fixing some gaps in the presentation of results.

Major comments:

- 1) Paper structure: the paper does not show a structure following the plan of Introduction, Results, Discussion, and Methods. It makes it difficult to follow. I would recommend making the structure more clear. I also think that a lot of the methods that are now in Supplementary Materials should be added to the main paper, and referenced to clearly when applicable, to help the reader understand the research better without having to open a separate file.
- 2) Paper structure: in the intro, lines 72 to 77, the authors detail their hypotheses in a certain order, but this order is not respected in the paper, as "interactions with other microbes", numbered 3) in the intro, is one of the first results with the section on hyphal highway. For clarity, I think the order of hypotheses in the intro should match the order in which the results are presented in the paper.
- 3) Supplementary Figure 1: I think the panels in Supp. Figure 1 should be called specifically in the paper, instead of generally referring to Supplementary Figure 1. For example, line 82, panel (a) of Supp. Fig 1 should be referred to specifically. I would appreciate if the authors could make these specific references to panels of Supp. Fig 1 throughout the paper, making sure that every panel of Supp. Fig1 is explicitly referred to at some point of the paper.
- 4) Supplementary figure 1 panel (a): I think this panel needs reworking. It is supposed to show the design of the chip used in the study, but given the current resolution, it is impossible to get an idea of what the micro-engineered channels look like. If there is an issue of scale for showing both the global scale of the chip (as in current panel), and the microchannels in each part of the chip, I would recommend that the authors provide zoomed-in sketches of the microchannel designs in each different part of the chip next to the existing global view. I think it is very important since the

chip is novel and the diverse structures appear throughout the paper in micrographs and videos.

5) Results section and Figure 1 and 2: the authors need to make clear what was the nutrient treatment for the images/data reported. Are they showing images from the initially air-filled chamber, of diH₂O filled chamber, etc...? It is not specified as far as I can see, and is a relevant information.

6) Figure 1: I would suggest to explicitly refer to the relevant supplementary video in the caption when describing each panel from (a) to (i) (c excluded), instead of referring to all the videos at the end. It will make it much easier to directly access the relevant video for each process described when reading the paper.

7) Figure 1 and associated results: are there any statistics on the number of occurrences of the processes showed in Figure 1? For example, was there a unique sighting of a nematode trapped by a hypha, or was such a sighting made multiple times on multiple chips? It would help giving more relevance to the results if these numbers were available.

8) Results statistics: the only indication of the number of samples for bar plots such as figure 2b can be found in the Supplementary, which is not satisfying in my opinion. I think the authors should put the number of samples studied in the text, and in the caption of the figures, when suitable. For example, for Fig. 2b, I would expect the number of diamonds quantified to be mentioned line 132 next to the p-value, and also mentioned in Figure 2 caption or on the figure itself. Same for figure 2c and figure 3.

9) I would suggest that in the last section from line 251 onwards, the authors recapitulate the results they obtained and discussed their impact, before moving on with the existing discussion on the merits of microfluidic chips for the future of research in the field. It will help the reader clearly understand what the study showed in a single location.

10) Supplementary videos: I find that these videos are very nice and interesting for the community; however I believe they would gain a lot with some further editing to help guide the viewer. Specifically, I think it would be valuable to add the following features to the existing videos:

- o A title slide, which describes in a few words what we are going to see and which links the video to the corresponding subpanel in figure 1. For example, for Supplementary Video 3, I would suggest a title along the lines of "Fungal highway for bacteria dispersal", and reference made to Figure 1 (d).

- o It might be valuable in the same title slide to add the corresponding sketch from the bottom of Figure 1, as it would help prepare the viewer to distinguish the different elements to be seen (hyphae, bacteria, microfauna, etc...). Or, in the video itself, the authors could insert arrows pointing at the different microbes with labels, as they did in Figure 1.

- o I would appreciate if after the title slide and before the video, the authors could show a sketch of the chip and point out the region of interest where the specific movie was recorded. From now it is difficult to know where in the chips each video was taken, which I think is a very relevant information to provide.

- o Overlaid with the videos, the authors should introduce a scale bar to give an idea of the scale to the viewer.

Minor comments:

11) Line 24/25 "The chip [...] corresponded well with the soil's matrix": I find this sentence unclear and I am not sure of what the authors mean. Do they mean that it integrated itself nicely with the soil matrix, and thus was successfully colonized? I would advise to rephrase.

12) Line 33/34 "Also the microbes influenced...": I think "The microbes also influenced..." is a better formulation.

13) Line 48 "hungry microorganisms": "hungry" does not sound very scientific for microorganisms. I would suggest using "foraging" or "starving" or "metabolically active".

14) Line 60 "miniscule": this is a typo, it is "minuscule".

15) Line 63 to 67 "As a result [...] remains fragmentary": This sentence is too long, and would benefit from being broken in two parts. I would break it so that "Our knowledge of soil trophic interactions also remains fragmentary." becomes an independent sentence.

16) Line 92 "To evaluate...": to improve clarity, I would suggest rephrasing as "To evaluate the effect of different nutritional conditions, we tested three treatments on four replicate chips each: chips were filled with deionized water, nutrient medium or were left empty." Please also specify the nutrient medium used.

17) Line 108 to 111: the authors sometimes use "adjacent soil", sometimes "surrounding soil", for designating the soil in contact with the chip inlets. For clarity, I would recommend to use a single expression and be consistent with it throughout the paper.

18) Line 117 "live": I would suggest to replace "live" by "dynamically"

19) Line 123: the reference for hyphae highway for bacteria reads " Figs 1d, 2c,f", however panel 2c is about water invasion, not bacteria. Did the authors mean "Figs 1d, 2b,e,f"?

20) Line 138: "Fig2c" should have a minuscule "c"

21) Line 146 to 150 "Bacterial abundance [...] bacteria themselves.": the sentence is unclear and too long. Please reformulate.

22) Line 164: When moving from nutrient selectivity to geometrical effect, please break the paragraph. It will make things clearer.

23) Line 178 to 180 "In addition to microorganisms [...] soil solution": "also" should be before the verb.

24) Line 185 "Preferential flow paths are the main water movement channels in soil": this expression is unclear. What do the authors mean? Do they mean that the preferential flow paths are responsible for the bulk of water transport in soil? If so please reformulate.

25) Line 204: Specify next to the reference to Supplementary videos 10 and 11 which one is the tracked particles, which one is the contrasted video, so that the reader can choose what he wants to see.

26) Line 220 "...resemble reality more closely.": "reality" sounds a little unscientific, I would suggest to reformulate, maybe as "make the inside of the chip more similar chemically to the natural environment".

27) Line 225: the reference for hyphae changing the pore space says "Fig 1e,g,i", however in Figure 1i there is no hyphae. Did the authors mean "Fig 1e,g,h", since panel h shows hyphae?

28) References: some of the references have dois, some do not have dois. I would recommend a uniform style, with or without dois.

29) Figure 1: caption for panel b says "Motile soil particles", when it should say "Mobile soil particles" since the soil particles do not swim by themselves but are moved by water currents.

Dear Reviewers,

Thank you cordially for your constructive and encouraging comments on our manuscript “Microfluidic chips provide visual access to *in situ* soil ecology”, where we study soil microbial dispersal and interactions with each other and the simulated soil pore space.

As suggested, we have now performed two new, major experiments in the laboratory: We used a whole-soil inoculum to study soil processes in the chips in a controlled and time-resolved manner in the lab, corroborating our results on the influence of microstructures and chip-filling conditions on microbial dispersal, and the influence of fungal hyphae on both bacterial dispersal, and even allows us to expand this finding to the dispersal of protists.

Combining *in situ* buried chips with soil-inoculated microcosms in the laboratory allows us to take advantage of both the possibility to study microbial processes embedded in their complex communities and realistic environmental conditions, with the careful control during laboratory experiments, allowing us to monitor processes over time. It also allows us, thanks to the high internal replication of measurement units within the chips, statistically combined with physical replicates of multiple chips, to draw relevant conclusions about the *relevance and frequency* of these processes.

In the following we address the individual comments (our responses marked in green):

Reviewer #1 (Remarks to the Author):

Recommendation: Major Revision

Overall impression:

This was an extremely interesting and fun paper to read. This work offers a unique perspective on the use of a ‘Soil on Chip’ platform and demonstrates how these types of platforms can be used to generate new hypotheses about multi-kingdom interactions. I applaud the authors for their bravery in attempting to bury these chips in a natural environment and then ‘see’ what happens. The authors clearly and honestly describe the limitations of the synthetic, 2D material platform, but make appropriate use of the different shapes in their model system to look the effects of confinement and connectivity on co-colonization and wetting.

Thank you for your encouraging reaction and enthusiasm, and we are glad you appreciate our approach!

I enjoyed reading the paper and found that it stimulated new ideas for my own work about how to test specific hypotheses that were posed, however it seemed to fall short in

truly enabling or executing a specific set of experiments that truly prove or disprove these hypotheses or delve into the mechanisms that drive the phenomena that were observed. As a collection of interesting observations that were made, enabled by this platform, I see value in publishing this work, though it is honestly difficult to gauge the value of the contribution. The concept and approach are notable, burying a discovery platform with lots of variables built into it underground and seeing what happens is brave and unique. However, the majority of the manuscript seems to be speculative and observational. Some quantitative analysis is done, most notably the difference in bacterial colonization and hyphal presence, but additional experimentation is warranted to give this manuscript true scientific rigor.

Given these observations, it would be worthwhile to replicate that phenomena in the lab using a known fungal species and small collection of soil isolates within a microfluidic chip that contains the diamond design. The dispersal experiment could and should be replicated in the lab in a similar manner. Are the results species dependent? Do subtle variations in the geometry (channel width for the zig zag, etc. impact dispersal)?

As you suggested we have now performed two new, major experiments in the lab in which we think we combine the advantages of both our first approach – burying the chips in a natural soil ecosystem (Now denoted Expt. 1)– with the advantages of controlled laboratory experiments. We collected real soil inoculum and used it in a controlled and time-resolved manner in the lab, which allows us to take advantage of both the possibility to study microbial processes embedded in their complex communities and to carefully control the experiments and measure them over time: Results lines. 122-138, 158-189, Figures 2 d-g, Fig. 4, Suppl. Fig 3. We investigated the influence of microstructures on microbial dispersal from soils taken into the laboratory where we could observe processes over time (Expt. 2; Fig. 2, Suppl. Fig. 3, Suppl. GIFs 1-3), and the influence of fungal hyphae on both bacterial and protist dispersal measured over time (Expt. 3, Fig. 4, Suppl. GIFs 4-6). All those approaches are completely novel and allow us to make data collections previously not possible. It also allows us, thanks to the high internal replication of measurement units within the chips, statistically combined with physical replicates of multiple chips, to draw relevant conclusions about the *relevance* of these processes and give us an estimate of their *frequency*. We hope that you agree with us that this significantly lifted the relevance of this paper.

Overall, I think that the approach of using microfluidics, buried in the environment, and then bringing them out to look at what happened is a novel notion for generating new hypotheses, but given the emergence of other soil chip platforms, and the missed opportunity to follow up on some of these observations with more targeted and controlled experiments, this manuscript falls a bit short with respect to the novelty and rigor necessary for publication at this time. I recommend performing a few follow up experiments to support the hypotheses generated using the platform and resubmitting.

Following your advice, we performed new experiments that are detailed above.

Additionally, addressing the more specific comments below is warranted.

More detailed comments:

1. The language/grammar needs some additional editing to improve the clarity and precision of the writing. I know that this is a bit picky, but some thorough editing and proofreading are needed. I've picked 3 examples below, but there were quite a few.

- a. Line 22, use of the word ‘got’ with ‘was’
- b. Line 25, use of the phrase ‘corresponded well..., allowing us to eavesdrop live and in a more realistic way...’ is a bit of an exaggeration. How did it correspond? Were sizes, gaps, similar to soil? What were you ‘eavesdropping’ on? Did you measure any chemical species that drove the observed chemical reactions? The observations were made in the lab after the chips were colonized, this isn’t truly done in the soil environment.
- c. Line 59, ‘organization at microscale’ should read ‘organization at the microscale’

We corrected language issues by professional proof reading. After performing the new experiments, we changed the abstract content and therefore the mentioned phrases a & b in line 22 and 25 do not longer exist.

c: corrected.

2. Lines 185-211 describe the use of the platform for observing preferential flow paths and potential for understanding solute transport and predicting ‘the way microbes react to and interact with different flow movements in the soil.’ -- recent work by Aufrecht et al. describes the development of a soil-on-a-chip platform that replicates the porosity and packing of soils and demonstrates the influence of pore scale hydrodynamics on bacterial attachment and alteration of flow. The discussion here should be modified to reflect this work and should be compared/contrasted appropriately.

Aufrecht, J. A., Fowlkes, J. D., Bible, A. N., Morrell-Falvey, J., Doktycz, M. J. & Retterer, S. T. Pore-scale hydrodynamics influence the spatial evolution of bacterial biofilms in a microfluidic porous network (2019). PloS one 14, e0218316)

We incorporated the work of Aufrecht et al., and compare our complementary approaches in lines 321-328: “Chip studies on time-resolved microbe-habitat interactions constitute a valuable addition to studies in 2-D thin sections of soil microbe distribution¹⁴, e.g., for investigations of bacterial biofilm settlement in relation to different shear forces in a simulated sand pore space⁴⁹. Aufrecht et al. (2019) successfully identified the spatial distribution of fluorescent bacterial strains when inhabiting a microfluidic chip with water flows around a fixed solid PDMS matrix, while we here use a pore space combining fixed and mobile solids (the PDMS structures and the soil particles, resp.) which allow us to record spatial changes in the pore space as a response of water flows.”

3. In Figure 2, panels e, f. There is no reason to fit a curve to those plots. Remove it.

Following your suggestion, we have removed the curves.

4. I did not understand the contingency diagram used in figure 2e. Perhaps there is a more straightforward way of expressing your point.

The contingency diagram is considered the best way to demonstrate the non-parametric statistics performed on categorical datasets. We added percentage axis labels to make the interpretation easier for readers not used to this: It shows the percentage of analyzed diamond-shaped pore widenings that contain a hypha on the x-axis versus the percentage

of widenings containing liquid instead of air (the initial condition). We now also explain this explicitly in the figure caption: “Contingency diagram showing the occurrence of air or water in diamond-shaped openings depending on the presence of a fungal hypha. Along the x-axis the frequency of channels containing hyphae, and along the y-axis the frequency of channels containing liquid or water, is shown, 264 observations in total.”

5. In general, I was confused about what data came from which chip, what were included in the replicates and statistics for the different figures. It would be helpful if some of this information were pulled out of the supplement and added, in brief, to the figures.

We added this information to all figure and video captions.

Reviewer #2 (Remarks to the Author):

Review Report: A window to the underground – micro-chip gives visual access to in-situ soil ecology

Mafla-Endara et al. describe a micro-engineered system that allows researchers to gain visual access to soil ecology *in situ*. Specifically, the authors placed micro-engineered chips, fabricated from the elastomeric polymer poly(dimethylsiloxane) (PDMS), into soil and left them there for a period of two months. They then dug out the chips and made observations regarding what had colonized the architecture, identifying a variety of soil organisms and processes taking place in the microchannels.

Although very simple in nature, this is – to the best of my knowledge - the first account that details an experimental setup whereby one has put soil onto a micro device, which can then be utilized to investigate soil ecological processes *in situ*. It is especially interesting that fungal highways have been observed *in situ* using real soils. This study highlights how lab-on-a-chip technology can help to elucidate (and begin to quantify) processes taking place in soil environments. It is refreshing to see that the authors also appreciate the limitations of the device setup, whilst at the same time informing of the potential future advantages.

We thank you for showing appreciation of our approach.

Major comments

1. The introduction needs to include an appreciation of how microchips have been used previously to study soil organisms using Lab-on-a-Chip technology, i.e. to put the study into context. This is lacking at the moment. The concept of Soil-on-a-Chip technology, i.e. the use of microfluidic technology to create soil-like environments to understand the complexity of dynamic below-ground biological processes at the microscale, with high spatio- and temporal resolution, is not new and should be discussed in more detail, including the appropriate references, e.g. Stanley et al. Lab Chip, 2016 (“Soil-on-a-Chip”) etc., as well as a reference to Nicolas et al. Appl Environ. Microbiol, 2010 (iChip, first chip to be put into soil; the authors reference this but it should be included in the introduction) etc.

We have now described previous studies including the ones mentioned above in the

introduction (lines 60-66): “Microfluidic chips have already demonstrated their usefulness in controlling and shaping micro-environments for the study of cell-to-cell interactions, and revolutionized biomedical research with, e.g., organ-on-a-chip devices¹⁹. Even within soil science and microbial ecology, chips have been used to address important questions^{20,21} such as how to increase the number of culturable bacteria from the environment²², how bacteria spatially organize in a pore space along chemical gradients²³, and how intracellular signals propagate in fungal networks²⁴.”

2. The exact chip design is not clear: Figure S1 a and b are very difficult to read and do not detail the exact features of the device (the photograph in b indicates there are many finer structures). In terms of providing scientific transparency and allowing experiments to be reproduced, this needs to be specified in more detail. A clear “to-scale” 2D drawing (including the exact microchannel structure), or alternatively the original CAD file, needs to be included. Also, the specific regions of the device design (A-E) should be enlarged so it is easy for the reader to understand the geometries used in microchannel design etc. Then important feature dimensions, scale bars etc. can be included to complement the description in the text. The central panel in Figure 1 goes some way to addressing these issues, but it is more artistic in nature and not an accurate representation of the exact chip design.

We added all relevant information about the chip design to Figure S1, including an overview and details from the AutoCAD files, and describe details in Material and Methods (lines 396-425): “We constructed a proxy of a soil pore space system in a micro-engineered silicone chip containing different sets of microstructures, termed the “Obstacle chip”²⁵. The chip consists of an artificial pore system open on one side for inoculum, and it was designed to investigate the growth and dispersal behaviour of soil microbes (**Supplementary Fig. 1a, b**). The chip’s pore-space dimensions were optimized to match the dimensions of fungal hyphae, with structure widths ranging between 4 and 100 μm , and a uniform height of approximately 7 μm to aid microscopy, since cells are located in the same focal plane and rarely overlay. It contained five different geometric sections accessible by soil microbes via a common entry area (**Supplementary Fig. 1a**). The entry area consists of an open area with round pillars of 100 μm diameter at a separation of 100 μm , holding up the chip’s ceiling. It was cut open longitudinally with a scalpel prior to bonding (see below, section ‘**Chip fabrication**’), ensuring direct contact of the soil with the chip’s interior. The inner section comprises a combination of differently shaped channels and obstacles constituting five experimental sections, of which two were systematically examined in this study: 1) Section C: A set of channels with sharp corners of three different types (n=12, randomly distributed): zigzag channels (90° turns with all channel sections at $\pm 45^\circ$ angle from the main growth direction), meandering square channels (90° turns with each section oriented in either the main growth direction or perpendicular to it), ‘z’-shaped channels (sharp corners diverting 135° from the previous growth direction, with channel sections in the main growth direction and at angles of 45° and 135° from it); 2) Section D: So-called ‘diamond channels,’ with a repeated combination of 10 μm wide and 400 μm long straight channels alternating with 140 μm wide diamond-shaped widenings. This channel type was replicated in 36 channels, each containing 33 diamond widenings. The widenings were used as quantification units to count bacteria and protist cell numbers, and for determination of liquid ingression, for the experiments on dispersal via fungal hyphae. Section A of the chip contained systems of hexagonal pillars of different diameters, Section B consisted of straight channels with different widths, and Section E contained

two types of obstacle courses comprised of complex structures. Those and the entrance system provided space for general observations. The design was constructed in AutoCad 2015 (Autodesk), in which patterns within experimental Sections A and C were randomized using a custom script from UrbanLISP (<http://www.urbanlisp.com>).”

3. The devices are implanted into the soil and then removed ca. 2 months later. How do you ensure that the environment is not altered upon (i) removal from the soil and (ii) transport to the lab? E.g. hyphae are very fragile and could easily be disturbed if there is movement of the soil adjacent to the device opening during transport.

We handled the chips with our highest possible care, and the burial site is located just outside the laboratory building so that transport could be performed by slowly walking. Further, the results we now obtained from the chips incubated with soil in the laboratory confirmed the results from the buried chips, which reassures us that no larger disturbances happened between excavation and analysis.

We now explain in more detail within Material and Methods (lines 475-481): “The chips were placed horizontally in the soil at a depth of 10 cm in which the PDMS chip was facing up and the glass slide down. Horizontal placement was chosen to probe a single stratum of the soil, serving as a comparable inoculum to the whole of the entry system, and to aid non-destructive recovery. The soil was carefully placed back in its original orientation, and the litter layer was placed back. A string attached to each chip was placed with its opposite end above the soil surface and attached to a pin, to guide future retrieval.”;

Lines 485-493: “Thus, after 64 days, the chips were collected by carefully removing soil around the string leading to each chip. We carefully kept the adjacent soil atop the glass slide along the opening of the chip, to keep our artificial pore system connected to the real soil pore system, and to avoid such disturbances as hyphal tearing or evaporation of the liquid inside the chips (**Supplementary Fig. 1d**). We cleaned the chip windows by softly wiping them with a clean wipe and deionized water. Samples were carefully transported to the microscopy facilities, located adjacent to the burial site. The chips were harvested one at a time and analysed under the microscope immediately after collection and cleaning.”

4. How did the authors justify the device design? Is there a logical / mathematical rationale behind the design, or is it just random? It is difficult to really understand / appreciate this as the full device architecture is not provided (see comment 2).

We now explain the chip design in much greater detail in Material and Methods (lines 396-425), and added an introduction to the rationale of the design to the end of the introduction, lines 73-88 :

”We asked whether the dispersal capability of three functional microbial groups—fungi, bacteria and protists—into a pristine pore space environment is influenced by pore space characteristics such as their geometric shape and chemical conditions, and by interactions with other microbes. We further examined how the microhabitats themselves are affected

by abiotic and biotic factors such as drying and rewetting of the soil, and by the microorganisms themselves. The chip design contained different experimental sections with distinct geometrical patterns²⁵ (**Supplementary Fig. 1a, Sections A-E**), which we used to address the following specific questions: a) How is microbial dispersal influenced by the pore spaces being filled with air, water, or nutrient medium? b) How does pore space geometry affect microbial dispersal, such as channels angled in zigzag patterns, forcing the microbes to navigate through increasingly sharper turns? c) Are bacteria and protists influenced in their dispersal capabilities to new pore spaces by the presence of a fungal hypha? and d) How does drying and rewetting soil, and the moving and growing microorganisms, affect the spatial arrangement of the chips' pore space?"

5. Figure 1 (f) refers to the fact that mycophagy is detected in this device. How can the authors prove this is mycophagy? There looks to be some debris surrounding the hypha in video 5, but can one say absolutely that this is debris that originated from the hypha and that bacteria are feeding on it?

We decided to remove this image and the statements around it, since we unfortunately cannot prove our case firmly enough. The hypha along the channel was intact except for the section recorded, where it was clearly broken and leaking. Unfortunately we did not record the hypha from the entrance of the channel where it was clearly shown that the debris did not come from the pillars entry but from the fungus itself. We also cannot say whether the hyphal leakage occurred first and attracted bacteria, or whether the bacteria caused the hyphal injury.

We exchanged this image in Fig. 1 to another at least as an interesting phenomenon: Hyphae opening up new passages by strong hyphal force (new Fig. 1d).

We are certain on the fact that hyphal turnover happens within the time frame investigated in this study (observed manyfold, see Supplementary GIF 3 as an example), and still want to show the video of bacteria feeding around the hyphae from the old version of Fig. 1 (now Supplementary Movie 15). We however put it now into a more general context (lines 225-228):

“The cells that died inside the chip laid the base for an initial organic matter build-up, and necromass inside the system was recycled within the food web which was clearly visible when bacteria accumulated around fractured cells and organisms (**Supplementary Movie 15, Supplementary GIF 3**).”

Also, the authors claim that a nematode is trapped by hyphae in Figure 1 (g). How can the authors prove that the nematode is indeed trapped? The video does not show that the nematode is unable to exit the box, it is sat immobile throughout the duration of the video (video 6). Equally plausible is that the hyphae were already there when the nematode entered. PDMS is an elastomeric polymer (i.e. flexible), and it would be relatively easy for the nematode to squeeze past hyphae. Interestingly, however, it seems that the nematode is feeding on the hypha in this video, which could be a point to include somewhere in the manuscript.

Hyphae frequently became obstacles that hinder the access of different soil organisms through the channels and also through the open pillar system. This becomes even more

plausible considering that the height of the chip is only 7 μ m, while the structures visible in the 2-D projection are at least 10 μ m wide and often more, and thus, squeezing past a hyphae might appear easier than it likely is. Hyphae, especially when several jointly grow through constrictions in the pore space, can completely block the open spaces, such as in Fig. 1e. We have observed nematodes trying and not, or hardly succeeding to get past a hypha in the chips numerous times, and now add an additional supplementary video from the open pillar system of the chip where nematodes struggle with the enmeshing hyphae in their surroundings. (Supplementary Movie 13). In this new manuscript version we avoid the absolute term “trapping” and use “obstruction” instead, to include those cases where passage is not completely impossible.

We write: “Hyphal colonization restricted microbial dispersal of larger organisms such as protists and nematodes by occupying and obstructing the access to free pore spaces (**Fig. 1g, Supplementary Movies 6, 13**). Both nematodes and larger protists were frequently observed having difficulties passing hyphae in the chip with its low height of 7 μ m, especially if the hyphae occupied narrow constrictions in the solid pore matrix of the chip.”, lines 215-220.

We consider this image to be great to convey this phenomenon to the reader given the large size of the nematode, and the complete sealing of all possible passages by the hypha, and would therefore like to keep it in Fig. 1, and hope that you agree us, upon this additional information.

The extensive observations we made for this nematode-fungal interaction in Fig 1 (beyond the video we show here) does not allow to claim that the nematode is feeding on the hypha. To study fungi embedded in the soil food web is however surely an exciting future possibility.

6. Please include for all figures what “n” and the error bars represent.

All statistically relevant information including n and error bars are now carefully stated in the figure captions.

7. Scale bars need to be added to each of the videos. I also find it unusual to include “©SoilChip” on each of the videos and feel this should be removed for a scientific manuscript.

We added a title and scale bars, and removed the watermark.

Minor comments

Line 41: “Soil” not “soils”

Changed.

Line 42: Reconsider English

Changed; we language-checked the complete manuscript again.

Line 47: should read “extraordinarily species-rich”

Changed.

Line 48: "...many hungry microorganisms." Please use more scientific terminology

Changed to "substrate-limited microorganisms".

Line 76: at the micrometer scale

Changed.

Line 92: add a reference to Held et al. Fungal Biol, 2011 and/or Stanley et al. Integr Biol, 2014 (first studies looking at fungal growth and interactions with bacteria in microchannels environments, which discuss the advantage of compartmentalization)

We have now incorporated both studies into the introduction.

Line 93: can the authors justify why they use malt medium over other media such as potato dextrose broth?

The choice was initially convenience since our microbial cultures grow successfully in this medium. We have not compared different media so far, and thus cannot state which nutrient medium would be best at the present time point – comparing different bait media would be a study of its own. We however can clearly state now that the presence of a nutrient medium *per se* in the chips increases bacterial cell numbers manyfold, and consecutively increases the number of bacterial feeder protists considerably, which is our main point here.

Line 97-98: "10 μm wide channels of different angles" – what does this mean exactly?

We now write in the introduction (lines 83-85): "...b) How does pore space geometry affect microbial dispersal, such as channels angled in zigzag patterns, forcing the microbes to navigate through increasingly sharper turns?"

We added much more detailed information on the chip design via Suppl. Fig 1, and as text to main manuscript in Material and Methods (lines 409-415):

"Section C: A set of channels with sharp corners of three different types (n=12, randomly distributed): zigzag channels (90° turns with all channel sections at $\pm 45^\circ$ angle from the main growth direction), meandering square channels (90° turns with each section oriented in either the main growth direction or perpendicular to it), z-shaped channels (sharp corners diverting 135° from the previous growth direction with channel sections in the main growth direction and at angles of 45° and 135° from it)"

Line 96-100: English needs rewording as it is not clear at present. Also, the authors should refer to a supplementary figure (see my major comment #2 above) in order to help the reader understand what they mean to convey exactly. It is a little confusing at present.

This comment concerned the approach for analysis of hyphal-bacterial interactions in the chips during Expt 1.

We now explain the chip experimental designs in great detail in the M&M section (lines 395-415), and explain the experimental procedure in detail (lines 498-504):

“To analyse the effect of fungal hyphae on bacterial abundance, we recorded real-time videos slowly scanning along the whole length of the diamond-shaped opening channels (each 33 diamonds, section D in Supplementary **Fig. 1a**; **Fig. 3**). The rather sparse hyphal colonization allowed us to select pairs of channels where in the first channel a hypha had proliferated far into the channel, combined with a directly adjacent channel without hyphae, n=4. We then counted the number of bacterial cells in diamond-shaped widenings, the presence or absence of fungal hyphae, and the presence or absence of liquid inside each diamond.”

Line 101: Why horizontally and not vertically, for example? Please provide a scientific justification for this.

We did not want to introduce extra parameters such as chip inclination.

We now explain in the Methods (lines 476-478): “Horizontal placement was chosen to probe a single stratum of the soil, serving as a comparable inoculum to the whole of the entry system, and to aid non-destructive recovery.”

Line 102: Why 64 days? Again, please provide a scientific justification for this time frame.

We now explain in the Methods (lines 483-486): “Preliminary experiments had shown that a two-month incubation period would grant the colonization of different types of soil microorganisms and minerals, and a stabilization of the inner environmental conditions between the soil chip and the surrounding soil. Thus, after 64 days...”

Line 103: “promptly examined under the microscope”. What does this mean? Same day? Were the samples stored or the temperature regulated to represent the external conditions? Perhaps the comparatively warm temperature of the lab cf. the belowground soil floor could influence the observed interactions?

We reformulated this phrase and explain in Material and Methods (lines 492-493): “The chips were harvested one at a time and analysed under the microscope immediately after collection and cleaning. “

We did not regulate the temperature during examination in the laboratory, but due to the prompt analysis we expected this not to have a large influence. After the analysis for data in Figures 1-3, “the chips were left uncovered at room temperature for 60 minutes to initiate air drying in the adjacent soil, in order to observe the real-time effects of drying on organisms and particles in the pore space system of the chips. The adjacent soil was re-wetted by adding 400 μ l of water”; lines 505-508.

Line 128-129: The authors mention that there are hypha-free channels directly adjacent to channels containing hyphae. How do they ensure that there are hypha-free channels, which can be directly compared to the hypha-occupying channels? This is unclear, as it is not possible to visualize in which channels these measurements were conducted. Is it just random? Or perhaps the hypha-free channels induce a bias, i.e. exclusion of bacteria due to channel architecture rather than absence of a hypha. This is not clear and needs to be addressed.

Channel architecture was exactly identical. Having 36 channel replicates and a sparse hyphal colonization, we were able to choose this paired design: channels that were adjacent to each other, one with and one without hypha. We now write in lines 500-502:

” The rather sparse hyphal colonization allowed us to select pairs of channels where in the first channel a hypha had proliferated far into the channel, combined with a directly adjacent channel without hyphae, n=4.”

Line 145-146: English needs revising

We removed this sentence.

Line 148: typo

Changed.

Line 158: Do the authors give a reason why fungal hyphae avoided chips with malt agar cf. chips filled with water/air? Do the authors know what fungal species are growing in their chips (perhaps this would give some insight as to why)? This is quite surprising, as other published work show that fungal hyphae grow readily in nutrient-filled microchannels (e.g. Ghanem et al. ES&T, 2019 and Stanley et al. Integr Biol, 2014)

We agree that this was surprising, and after much more experimentation experience we removed this statement. We see a very variable colonization of fungi from soil inocula in the different chip replicates irrespective of chip filling, and random priority effects might play a large role. We now write:

“... fungal hyphae showed variable results, with strong or weak growth without a consistent effect of chip filling in both the field- and lab incubated settings.” (lines 120-121);

We also now discuss more thoroughly in lines 248-252:

“The variation found in our results suggests however that fungal reactions to microspaces can depend on factors not specifically tested in this study, such as priority effects by the order in which individual species entered the channels²⁵, or seasonal variation. It has been suggested that fungal hyphae prefer growing in air-filled pore spaces²⁸, but our study did not confirm this to be generally true.”

Line 161: Please show the data, at least some representative microscopic overviews if possible. This is very interesting and any scientific statements, such as this, should be backed up with data.

This comment is related to the comment above and specifically referred to hyphal colonization of the chips regarding the chip filling, the statement that the strongest hyphal colonization was found in the air-filled chips. After adding several more datasets this statement cannot be clearly supported anymore – we found variable fungal colonization, high or low irrespective of the filling. Therefore, we decided to delete it; see also reasoning above.

Line 400-401: I do not understand what the authors mean with the terminology “water retention bridges”, please explain.

We refer to the viscous water films that can connect solid parts of the soil pore space under semi-dry conditions, and increasingly form under the influence of dissolved organic compounds such as microbial secreted substances (e.g. EPS). We changed this term now to: “Water meniscus connecting soil particles and chip structures”, line 749, which is a term commonly used in soil science.

Supplementary Information: Methods

Line 4: photograph

Changed.

Line 46: specify glass thickness and supplier information

We added to Material and Methods (lines 444-445): “Twelve glass slides sized 55x75 mm and 1 mm thick (Thermo Scientific)...”

Line 68: What is the malt medium exactly?

We added to Material and Methods (lines 457-460): “... liquid malt medium, a complex medium to provide a nutrient-rich environment including reduced sugars such as disaccharide maltose and in lower proportion nitrogenous components such as peptides, amino acids purines and vitamins (malt extract for microbiology, Merck KGaA)...”

Review of “A window to the underground: microchip gives visual access to in-situ soil ecology”

Mafla-Endara et al.

In this paper, the authors present results related to the microbial colonization of a microchip which represent a model for empty spaces in soil, designed to encompass different complex structures mimicking potential features of soil. After 2 months of exposure to the natural environment, the micro-chips were recovered and observed mainly with optical microscopy to observe how different microbial groups had invaded its structure. A wide variety of microbes was observed in the chip, together with inert soil particles which also invaded the structure. Together microbes and particles created new microenvironments. The paper also provides some results on the dispersal of bacteria along fungal hyphae, the preference on dry/wet substrate for different microbial groups, and observations on preferential path flows inside the colonized chip.

I enjoyed reading this paper and it is relevant for the field of microbial ecology in soil. While PDMS chips have been used widely in other biological fields, the particular chip used here and its in-situ deployment in soil for observing natural assemblages is novel. I find the multiple observations of natural communities inside the chip and the videos provided as supplementary very exciting and inspirational for future research, as they raise many questions on microbial interactions in soil. As such, this paper would surely stimulate further research into the topic.

We thank you for your kind words and appreciation of our approach.

However, the weaknesses of the paper are in my opinion that it provides more of a collection of observations without directly answering a strong scientific question. The results related to bacterial dispersal along hyphae and preference of microbes for dry/wet substrate are sound but have already been demonstrated in other contexts, as mentioned by the authors themselves. The results on preferential path flow in structured environment seem quite preliminary, with only one example of flow visualization using particle tracking, and no real quantitative estimation of the impact of the measured speed on the transport of a diffusive nutrient. This flow visualization thus looks like a demonstration of feasibility to me.

We agree that the paper initially focused too much on novel observations that we were very enthusiastic about. We however now went through our aims and questions and formulated them much more explicitly and systematically. We performed two new, major experiments in the lab, for which we collected soil inoculum and used it in a controlled and time-resolved manner in the lab, corroborating our first findings from the *in situ* experiments. We think we combine the advantages of our first approach – burying the chips in a natural soil ecosystem, taking advantage to study microbial processes embedded in their complex communities – with the advantages of controlled laboratory experiments, carefully controlling the experiments and measure them over time. We present our novel results in this revised version of the manuscript in lines. 122-138, 158-189, Figures 2 d-g, Fig. 4, Supplementary Figures 3 and 4. We investigated the influence of microstructures on microbial dispersal from soils taken into the laboratory (Fig. 2), the time-resolved succession in which the large microbial groups enter the chips (Suppl. Fig. 3), and the influence of fungal hyphae on both bacterial and protist dispersal (Fig 4, Suppl. Fig. 4). All those approaches are completely novel and allow us to make data collections previously not possible. It also allows us, thanks to the high replication of measurement units within the chips and statistically combined with physical replicates of multiple chips, to draw relevant conclusions about the *relevance* of these processes and estimate their *frequency*. We hope that you agree with us that this significantly lifted the relevance of this paper and made it a versatile and inspiring example for the research field.

I now move on to more specific comments on the paper which aim at improving clarity, readability and fixing some gaps in the presentation of results.

Major comments:

1) Paper structure: the paper does not show a structure following the plan of Introduction, Results, Discussion, and Methods. It makes it difficult to follow. I would recommend making the structure more clear. I also think that a lot of the methods that are now in Supplementary Materials should be added to the main paper, and referenced to clearly when applicable, to help the reader understand the research better without having to open a separate file.

This was because it originally was formatted for another journal and internally transferred to Communications Biology. We now changed the paper structure and formatting to match CommsBio's requirements.

2) Paper structure: in the intro, lines 72 to 77, the authors detail their hypotheses in a certain order, but this order is not respected in the paper, as “interactions with other microbes”, numbered 3) in the intro, is one of the first results with the section on hyphal highway. For clarity, I think the order of hypotheses in the intro should match the order in which the results are presented in the paper.

We refined our hypotheses, added new ones to the new experiments, and follow up them as categories in the results section and in the discussion:

Hypotheses:

“a) How is microbial dispersal influenced by the pore spaces being filled with air, water, or nutrient medium? b) How does pore space geometry affect microbial dispersal, such as channels angled in zigzag patterns, forcing the microbes to navigate through increasingly sharper turns? c) Are bacteria and protists influenced in their dispersal capabilities to new pore spaces by the presence of a fungal hypha? and d) How does drying and rewetting soil, and the moving and growing microorganisms, affect the spatial arrangement of the chips’ pore space?” Lines 82-88.

Results order:

- Pore space geometrical characteristics and filling (Hypotheses a and b)
- The influence of fungal hyphae on microbial dispersal (Hypothesis c)
- Microhabitat formation (Hypothesis d)

3) Supplementary Figure 1: I think the panels in Supp. Figure 1 should be called specifically in the paper, instead of generally referring to Supplementary Figure 1. For example, line 82, panel (a) of Supp. Fig 1 should be referred to specifically. I would appreciate if the authors could make these specific references to panels of Supp. Fig 1 throughout the paper, making sure that every panel of Supp. Fig1 is explicitly referred to at some point of the paper.

We now specifically refer to the different panels in Suppl. Fig 1, and all panels are mentioned specifically.

4) Supplementary figure 1 panel (a): I think this panel needs reworking. It is supposed to show the design of the chip used in the study, but given the current resolution, it is impossible to get an idea of what the micro-engineered channels look like. If there is an issue of scale for showing both the global scale of the chip (as in current panel), and the microchannels in each part of the chip, I would recommend that the authors provide zoomed-in sketches of the microchannel designs in each different part of the chip next to the existing global view. I think it is very important since the chip is novel and the diverse structures appear throughout the paper in micrographs and videos.

We agree. We added all relevant information about the chip design to Figure S1, including an overview and details from the autoCAD files. We now explain the chip design in much greater detail in Material and Methods (lines 395-425).

5) Results section and Figure 1 and 2: the authors need to make clear what was the nutrient treatment for the images/data reported. Are they showing images from the initially air-filled chamber, of diH₂O filled chamber, etc...? It is not specified as far as I can see, and is a relevant information.

This information is now added to the Figure captions.

6) Figure 1: I would suggest to explicitly refer to the relevant supplementary video in the caption when describing each panel from (a) to (i) (c excluded), instead of referring to all the videos at the end. It will make it much easier to directly access the relevant video for each process described when reading the paper.

We now refer to each supplementary video explicitly where relevant.

7) Figure 1 and associated results: are there any statistics on the number of occurrences of the processes showed in Figure 1? For example, was there a unique sighting of a nematode trapped by a hypha, or was such a sighting made multiple times on multiple chips? It would help giving more relevance to the results if these numbers were available.

Each reported observation of Fig. 1 was recorded at least 3 times, and one sentence referring to this is included in the caption of Fig.1 (line 759).

8) Results statistics: the only indication of the number of samples for bar plots such as figure 2b can be found in the Supplementary, which is not satisfying in my opinion. I think the authors should put the number of samples studied in the text, and in the caption of the figures, when suitable. For example, for Fig. 2b, I would expect the number of diamonds quantified to be mentioned line 132 next to the p-value, and also mentioned in Figure 2 caption or on the figure itself. Same for figure 2c and figure 3.

We now explain the number of replicates explicitly in each experiment in Material and Methods and also added this information in the figure captions and to the description of the statistical results where relevant.

9) I would suggest that in the last section from line 251 onwards, the authors recapitulate the results they obtained and discussed their impact, before moving on with the existing discussion on the merits of microfluidic chips for the future of research in the field. It will help the reader clearly understand what the study showed in a single location.

We added a brief recapitulation at the beginning of the discussion (that is not overlapping with the brief results summary at the end of the discussion which is required by the Communications Biology format):

"The soil chips, colonized by a rich microbial community, constitute literal windows into the soil, allowing us to monitor soil processes in real time, and multi-level interactions among and between microorganisms and their habitat (**Fig. 1**). We examined the effect of principal pore space characteristics (**Fig. 2**) and showed that fungal hyphae frequently increased pore connectivity and dispersal of water dwelling organisms (**Figs. 3, 4**), which is likely of high relevance for their colonization success in an ever-changing pore space

system. We demonstrated that the dynamics of changing microhabitats are caused by both physical forces and biological activity (Fig. 5).”, lines 235-242.

10) Supplementary videos: I find that these videos are very nice and interesting for the community; however I believe they would gain a lot with some further editing to help guide the viewer. Specifically, I think it would be valuable to add the following features to the existing videos:

- o A title slide, which describes in a few words what we are going to see and which links the video to the corresponding subpanel in figure 1. For example, for Supplementary Video 3, I would suggest a title along the lines of “Fungal highway for bacteria dispersal”, and reference made to Figure 1 (d).

- o It might be valuable in the same title slide to add the corresponding sketch from the bottom of Figure 1, as it would help prepare the viewer to distinguish the different elements to be seen (hyphae, bacteria, microfauna, etc...). Or, in the video itself, the authors could insert arrows pointing at the different microbes with labels, as they did in Figure 1.

- o I would appreciate if after the title slide and before the video, the authors could show a sketch of the chip and point out the region of interest where the specific movie was recorded. From now it is difficult to know where in the chips each video was taken, which I think is a very relevant information to provide.

- o Overlaid with the videos, the authors should introduce a scale bar to give an idea of the scale to the viewer.

Nice to hear that you appreciated the videos. We now incorporated a title, the sketch of the capture including legend into each video, placed left of the footage during the whole run of the video, and added a scale bar.

Minor comments:

11) Line 24/25 “The chip [...] corresponded well with the soil’s matrix”: I find this sentence unclear and I am not sure of what the authors mean. Do they mean that it integrated itself nicely with the soil matrix, and thus was successfully colonized? I would advise to rephrase.

We strongly restructured the abstract – with the new data, these aspects were not prioritized any longer and thus omitted.

12) Line 33/34 “Also the microbes influenced...”: I think “The microbes also influenced...” is a better formulation.

This sentence does not exist any longer in this form, and language was corrected by a professional service.

13) Line 48 “hungry microorganisms”: “hungry” does not sound very scientific for microorganisms. I would suggest using “foraging” or “starving” or “metabolically active”.

We now write “substrate-limited microorganisms”.

14) Line 60 “miniscule”: this is a typo, it is “minuscule”.

Changed.

15) Line 63 to 67 “As a result [...] remains fragmentary”: This sentence is too long, and would benefit from being broken in two parts. I would break it so that “Our knowledge of soil trophic interactions also remains fragmentary.” becomes an independent sentence.

We simplified the sentence, now stating “the capacity for addressing fundamental knowledge gaps in the field of soil science has been limited”, followed by a list of examples.

16) Line 92 “To evaluate...”: to improve clarity, I would suggest rephrasing as “To evaluate the effect of different nutritional conditions, we tested three treatments on four replicate chips each: chips were filled with deionized water, nutrient medium or were left empty.” Please also specify the nutrient medium used.

Since we now have a proper Material and Methods-section in the main manuscript we shortened the summary of the methods at the end of the introduction. This information can now be found in detail in lines 465-460:

“The chips were filled with one of the following three treatments: (1) deionized water, (2) liquid malt medium, a complex medium to provide a nutrient-rich environment including reduced sugars such as disaccharide maltose and in lower proportion nitrogenous components such as peptides, amino acids purines and vitamins (malt extract for microbiology, Merck KGaA), or (3) were left empty = air filled.”

17) Line 108 to 111: the authors sometimes use “adjacent soil”, sometimes “surrounding soil”, for designating the soil in contact with the chip inlets. For clarity, I would recommend to use a single expression and be consistent with it throughout the paper.

We now consistently use “adjacent”.

18) Line 117 “live”: I would suggest to replace “live” by “dynamically”

We reformulated this as: “The transparency of the chips allowed us to observe the primary colonization of a pristine pore space and soil microbial interactions in real time” (lines 105-107).

19) Line 123: the reference for hyphae highway for bacteria reads “ Figs 1d, 2c,f”, however panel 2c is about water invasion, not bacteria. Did the authors mean “Figs 1d, 2b,e,f”?

We now make sure that all figure panels are correctly referred to. This exact sentence does not exist in this form any longer as we included more data to this section.

20) Line 138: “Fig2c” should have a minuscule “c”

Changed.

21) Line 146 to 150 “Bacterial abundance [...] bacteria themselves.”: the sentence is unclear and too long. Please reformulate.

We split the sentence and reformulated to ” Bacterial abundance is known to be generally higher in the mycosphere than in bulk soil³². Laboratory experiments have shown how mycosphere bacteria are likely to have greater access to nutrients^{36–38}, both in the form of exudates and nutrients released during fungal degradation of organic matter..”, lines 272-276.

22) Line 164: When moving from nutrient selectivity to geometrical effect, please break the paragraph. It will make things clearer.

Agreed and changed. (L. X).

23) Line 178 to 180 “In addition to microorganisms [...] soil solution”: “also” should be before the verb.

Changed.

24) Line 185 “Preferential flow paths are the main water movement channels in soil”: this expression is unclear. What do the authors mean? Do they mean that the preferential flow paths are responsible for the bulk of water transport in soil? If so please reformulate.

We reformulated to ” Preferential flow paths are the main channels for water movement in soil,... “.

25) Line 204: Specify next to the reference to Supplementary videos 10 and 11 which one is the tracked particles, which one is the contrasted video, so that the reader can choose what he wants to see.

We added this information (line 204).

26) Line 220 “...resemble reality more closely.”: “reality” sounds a little unscientific, I would suggest to reformulate, maybe as “make the inside of the chip more similar chemically to the natural environment”.

We reformulated to: “...allowing organic coatings to form on the chip surfaces.”

27) Line 225: the reference for hyphae changing the pore space says “Fig 1e,g,i”, however in Figure 1i there is no hyphae. Did the authors mean “Fig 1e,g,h”, since panel h shows hyphae?

Changed, we now make sure to refer to the correct panels.

28) References: some of the references have dois, some do not have dois. I would recommend a uniform style, with or without dois.

We double-checked all references to match correct CommBiol- style.

29) Figure 1: caption for panel b says “Motile soil particles”, when it should say “Mobile soil particles” since the soil particles do not swim by themselves but are moved by water currents.

Changed.

REVIEWERS' COMMENTS:

Reviewer #1 (Remarks to the Author):

I appreciate the authors' efforts to address my comments. The performance of lab experiments to support observations and address hypotheses from the in situ experiments was a substantial effort and I applaud their efforts. I am satisfied with the revised manuscript and recommend publication.

Reviewer #2 (Remarks to the Author):

Review Report: A window to the underground – micro-chip gives visual access to in-situ soil ecology

Mafla-Endara et al. describe a micro-engineered system that allows researchers to gain visual access to soil ecology in situ. Specifically, the authors placed micro-engineered chips, fabricated from the elastomeric polymer poly(dimethylsiloxane) (PDMS), into soil and left them there for a period of two months. They then dug out the chips and made observations regarding what had colonized the architecture, identifying a variety of soil organisms and processes taking place in the microchannels.

Although very simple in nature, this is – to the best of my knowledge - the first account that details an experimental setup whereby one has put soil onto a micro device, which can then be utilized to investigate soil ecological processes in situ. It is especially interesting that fungal highways have been observed in situ using real soils. This study highlights how lab-on-a-chip technology can help to elucidate (and begin to quantify) processes taking place in soil environments. It is refreshing to see that the authors also appreciate the limitations of the device setup, whilst at the same time informing of the potential future advantages.

We thank you for showing appreciation of our approach.

The authors have addressed all of my comments adequately; however, I have a very minor additional request for Point 2 (see Major Comments section below in red).

In light of the new material that has been included in the manuscript, I have a few additional minor comments:

- The movies have been greatly improved; I really like them now. However, I was a little confused at first though by the key/legend, as I was expecting hyphae etc to be present in Movie 1 for example. As a suggestion, I think it would be more reader-friendly to include only those elements in the key/legend that are displayed in the cartoon overview (e.g. chip, soil, air, water for Movie 1).
- Please include the cartoon overviews in all movies (only Movies 1-8 have this).
- Is Movie 4 (Habitat fragmentation) actually a movie? I do not see anything happening (at least it is not obvious viewing it on my monitor).
- Each GIF needs to include a scale bar and a title (so that it is easy for the reader to grasp / remind them what it represents). The GIFs are a nice idea, however it is difficult to check both the time point and what is changing in the channels from frame to frame. Please include a Supplementary Figure for each GIF showing a time series (essentially each frame in the GIF side by side in a figure) with a description of what is happening. A description in the main manuscript is lacking and it is therefore not clear what each GIF is depicting.
- Fig 3C: please label top axis. Does this refer to the percentage of channels that either contain or do not contain hyphae?
- Line 804: revise English “was kept equable”
- Line 221: “by breaking up solid structures”. Please rephrase this sentence. Sometimes PDMS does not always bond properly/uniformly to glass. Therefore, in such regions you will see the hyphae squeezing between the PDMS and glass layers. I very much doubt

that the hyphae are breaking the covalent bonds formed between the glass and PDMS during plasma bonding (unless you can prove this?) and you would have major problems everywhere in your device if this was the case.

- I find it very confusing as to what n refers to exactly, i.e. when the authors describe each Experiment (lines 465 onwards). For example, in line 466 – 467 three filling treatments are described (deionised water, malt extract, air), and ‘n’ is said to equal 3. What is n here: 1 replicate for each condition, or 3 replicates of each condition. This needs to be clarified throughout the manuscript.

In summary, I applaud the authors on improving their study greatly with the addition of the new experiments and look forward to seeing this manuscript published in *Communications Biology*.

Major comments

1. The introduction needs to include an appreciation of how microchips have been used previously to study soil organisms using Lab-on-a-Chip technology, i.e. to put the study into context. This is lacking at the moment. The concept of Soil-on-a-Chip technology, i.e. the use of microfluidic technology to create soil-like environments to understand the complexity of dynamic below-ground biological processes at the microscale, with high spatio- and temporal resolution, is not new and should be discussed in more detail, including the appropriate references, e.g. Stanley et al. *Lab Chip*, 2016 (“Soil-on-a-Chip”) etc., as well as a reference to Nicolas et al. *Appl Environ. Microbiol*, 2010 (iChip, first chip to be put into soil; the authors reference this but it should be included in the introduction) etc.

We have now described previous studies including the ones mentioned above in the introduction (lines 60-66): “Microfluidic chips have already demonstrated their usefulness in controlling and shaping micro-environments for the study of cell-to-cell interactions, and revolutionized biomedical research with, e.g., organ-on-a-chip devices¹⁹. Even within soil science and microbial ecology, chips have been used to address important questions^{20,21} such as how to increase the number of culturable bacteria from the environment²², how bacteria spatially organize in a pore space along chemical gradients²³, and how intracellular signals propagate in fungal networks²⁴.”

2. The exact chip design is not clear: Figure S1 a and b are very difficult to read and do not detail the exact features of the device (the photograph in b indicates there are many finer structures). In terms of providing scientific transparency and allowing experiments to be reproduced, this needs to be specified in more detail. A clear “to-scale” 2D drawing (including the exact microchannel structure), or alternatively the original CAD file, needs to be included. Also, the specific regions of the device design (A-E) should be enlarged so it is easy for the reader to understand the geometries used in microchannel design etc. Then important feature dimensions, scale bars etc. can be included to complement the description in the text. The central panel in Figure 1 goes some way to addressing these issues, but it is more artistic in nature and not an accurate representation of the exact chip design.

We added all relevant information about the chip design to Figure S1, including an overview and details from the AutoCAD files, and describe details in Material and Methods (lines 396-425): “We constructed a proxy of a soil pore space system in a micro-engineered silicone chip containing different sets of microstructures, termed the

“Obstacle chip”²⁵. The chip consists of an artificial pore system open on one side for inoculum, and it was designed to investigate the growth and dispersal behaviour of soil microbes (Supplementary Fig. 1a, b). The chip’s pore-space dimensions were optimized to match the dimensions of fungal hyphae, with structure widths ranging between 4 and 100 μm , and a uniform height of approximately 7 μm to aid microscopy, since cells are located in the same focal plane and rarely overlay. It contained five different geometric sections accessible by soil microbes via a common entry area (Supplementary Fig. 1a). The entry area consists of an open area with round pillars of 100 μm diameter at a separation of 100 μm , holding up the chip’s ceiling. It was cut open longitudinally with a scalpel prior to bonding (see below, section ‘Chip fabrication’), ensuring direct contact of the soil with the chip’s interior. The inner section comprises a combination of differently shaped channels and obstacles constituting five experimental sections, of which two were systematically examined in this study: 1) Section C: A set of channels with sharp corners of three different types (n=12, randomly distributed): zigzag channels (90° turns with all channel sections at $\pm 45^\circ$ angle from the main growth direction), meandering square channels (90° turns with each section oriented in either the main growth direction or perpendicular to it), ‘z’-shaped channels (sharp corners diverting 135° from the previous growth direction, with channel sections in the main growth direction and at angles of 45° and 135° from it); 2) Section D: So-called ‘diamond channels,’ with a repeated combination of 10 μm wide and 400 μm long straight channels alternating with 140 μm wide diamond-shaped widenings. This channel type was replicated in 36 channels, each containing 33 diamond widenings. The widenings were used as quantification units to count bacteria and protist cell numbers, and for determination of liquid ingression, for the experiments on dispersal via fungal hyphae. Section A of the chip contained systems of hexagonal pillars of different diameters, Section B consisted of straight channels with different widths, and Section E contained two types of obstacle courses comprised of complex structures. Those and the entrance system provided space for general observations. The design was constructed in AutoCad 2015 (Autodesk), in which patterns within experimental Sections A and C were randomized using a custom script from UrbanLISP (<http://www.urbanlisp.com>).”

I appreciate that the authors have made a very good effort to improve the level of detail regarding the chip design. However, some information is still lacking and needs clarifying.

Section C: Please include microchannel width here too (so that readers have all information easily accessible in one place). It is also not possible to determine how long the channels are before the channel changes direction from this drawing.

Section E: This is still far too small

A scale bar needs to be included with each of the enlarged sections A-E

3. The devices are implanted into the soil and then removed ca. 2 months later. How do you ensure that the environment is not altered upon (i) removal from the soil and (ii) transport to the lab? E.g. hyphae are very fragile and could easily be disturbed if there is movement of the soil adjacent to the device opening during transport.

We handled the chips with our highest possible care, and the burial site is located just outside the laboratory building so that transport could be performed by slowly walking. Further, the results we now obtained from the chips incubated with soil in the laboratory

confirmed the results from the buried chips, which reassures us that no larger disturbances happened between excavation and analysis.

We now explain in more detail within Material and Methods (lines 475-481): “The chips were placed horizontally in the soil at a depth of 10 cm in which the PDMS chip was facing up and the glass slide down. Horizontal placement was chosen to probe a single stratum of the soil, serving as a comparable inoculum to the whole of the entry system, and to aid non-destructive recovery. The soil was carefully placed back in its original orientation, and the litter layer was placed back. A string attached to each chip was placed with its opposite end above the soil surface and attached to a pin, to guide future retrieval.”;

Lines 485-493: “Thus, after 64 days, the chips were collected by carefully removing soil around the string leading to each chip. We carefully kept the adjacent soil atop the glass slide along the opening of the chip, to keep our artificial pore system connected to the real soil pore system, and to avoid such disturbances as hyphal tearing or evaporation of the liquid inside the chips (Supplementary Fig. 1d). We cleaned the chip windows by softly wiping them with a clean wipe and deionized water. Samples were carefully transported to the microscopy facilities, located adjacent to the burial site. The chips were harvested one at a time and analysed under the microscope immediately after collection and cleaning.”

4. How did the authors justify the device design? Is there a logical / mathematical rationale behind the design, or is it just random? It is difficult to really understand / appreciate this as the full device architecture is not provided (see comment 2).

We now explain the chip design in much greater detail in Material and Methods (lines 396-425), and added an introduction to the rationale of the design to the end of the introduction, lines 73-88 :

”We asked whether the dispersal capability of three functional microbial groups—fungi, bacteria and protists—into a pristine pore space environment is influenced by pore space characteristics such as their geometric shape and chemical conditions, and by interactions with other microbes. We further examined how the microhabitats themselves are affected by abiotic and biotic factors such as drying and rewetting of the soil, and by the microorganisms themselves. The chip design contained different experimental sections with distinct geometrical patterns²⁵ (Supplementary Fig. 1a, Sections A-E), which we used to address the following specific questions: a) How is microbial dispersal influenced by the pore spaces being filled with air, water, or nutrient medium? b) How does pore space geometry affect microbial dispersal, such as channels angled in zigzag patterns, forcing the microbes to navigate through increasingly sharper turns? c) Are bacteria and protists influenced in their dispersal capabilities to new pore spaces by the presence of a fungal hypha? and d) How does drying and rewetting soil, and the moving and growing microorganisms, affect the spatial arrangement of the chips’ pore space?”

5. Figure 1 (f) refers to the fact that mycophagy is detected in this device. How can the authors prove this is mycophagy? There looks to be some debris surrounding the hypha in video 5, but can one say absolutely that this is debris that originated from the hypha and that bacteria are feeding on it?

We decided to remove this image and the statements around it, since we unfortunately cannot prove our case firmly enough. The hypha along the channel was intact except for the section recorded, where it was clearly broken and leaking. Unfortunately we did not record the hypha from the entrance of the channel where it was clearly shown that the debris did not come from the pillars entry but from the fungus itself. We also cannot say whether the hyphal leakage occurred first and attracted bacteria, or whether the bacteria caused the hyphal injury.

We exchanged this image in Fig. 1 to another at least as an interesting phenomenon: Hyphae opening up new passages by strong hyphal force (new Fig. 1d).

We are certain on the fact that hyphal turnover happens within the time frame investigated in this study (observed manyfold, see Supplementary GIF 3 as an example), and still want to show the video of bacteria feeding around the hyphae from the old version of Fig. 1 (now Supplementary Movie 15). We however put it now into a more general context (lines 225-228):

“The cells that died inside the chip laid the base for an initial organic matter build-up, and necromass inside the system was recycled within the food web which was clearly visible when bacteria accumulated around fractured cells and organisms (Supplementary Movie 15, Supplementary GIF 3).”

Also, the authors claim that a nematode is trapped by hyphae in Figure 1 (g). How can the authors prove that the nematode is indeed trapped? The video does not show that the nematode is unable to exit the box, it is sat immobile throughout the duration of the video (video 6). Equally plausible is that the hyphae were already there when the nematode entered. PDMS is an elastomeric polymer (i.e. flexible), and it would be relatively easy for the nematode to squeeze past hyphae. Interestingly, however, it seems that the nematode is feeding on the hypha in this video, which could be a point to include somewhere in the manuscript.

Hyphae frequently became obstacles that hinder the access of different soil organisms through the channels and also through the open pillar system. This becomes even more plausible considering that the height of the chip is only 7 μ m, while the structures visible in the 2-D projection are at least 10 μ m wide and often more, and thus, squeezing past a hyphae might appear easier than it likely is. Hyphae, especially when several jointly grow through constrictions in the pore space, can completely block the open spaces, such as in Fig. 1e. We have observed nematodes trying and not, or hardly succeeding to get past a hypha in the chips numerous times, and now add an additional supplementary video from the open pillar system of the chip where nematodes struggle with the enmeshing hyphae in their surroundings. (Supplementary Movie 13). In this new manuscript version we avoid the absolute term “trapping” and use “obstruction” instead, to include those cases where passage is not completely impossible.

We write: “Hyphal colonization restricted microbial dispersal of larger organisms such as protists and nematodes by occupying and obstructing the access to free pore spaces (Fig. 1g, Supplementary Movies 6, 13). Both nematodes and larger protists were frequently observed having difficulties passing hyphae in the chip with its low height of 7 μ m, especially if the hyphae occupied narrow constrictions in the solid pore matrix of the chip.”, lines 215-220.

We consider this image to be great to convey this phenomenon to the reader given the large size of the nematode, and the complete sealing of all possible passages by the hypha, and would therefore like to keep it in Fig. 1, and hope that you agree us, upon this additional information.

The extensive observations we made for this nematode-fungal interaction in Fig 1 (beyond the video we show here) does not allow to claim that the nematode is feeding on the hypha. To study fungi embedded in the soil food web is however surely an exciting future possibility.

6. Please include for all figures what “n” and the error bars represent.

All statistically relevant information including n and error bars are now carefully stated in the figure captions.

7. Scale bars need to be added to each of the videos. I also find it unusual to include “SoilChip” on each of the videos and feel this should be removed for a scientific manuscript.

We added a title and scale bars, and removed the watermark.

Minor comments

Line 41: “Soil” not “soils”

Changed.

Line 42: Reconsider English

Changed; we language-checked the complete manuscript again.

Line 47: should read “extraordinarily species-rich” Changed.

Line 48: “...many hungry microorganisms.” Please use more scientific terminology Changed to “substrate-limited microorganisms”.

Line 76: at the micrometer scale

Changed.

Line 92: add a reference to Held et al. Fungal Biol, 2011 and/or Stanley et al. Integr Biol, 2014 (first studies looking at fungal growth and interactions with bacteria in microchannels environments, which discuss the advantage of compartmentalization)

We have now incorporated both studies into the introduction.

Line 93: can the authors justify why they use malt medium over other media such as potato dextrose broth?

The choice was initially convenience since our microbial cultures grow successfully in this medium. We have not compared different media so far, and thus cannot state which nutrient medium would be best at the present time point – comparing different bait media would be a study of its own. We however can clearly state now that the presence of a nutrient medium per se in the chips increases bacterial cell numbers manyfold, and consecutively increases the number of bacterial feeder protists considerably, which is our main point here.

Line 97-98: “10 µm wide channels of different angles” – what does this mean exactly?

We now write in the introduction (lines 83-85): "...b) How does pore space geometry affect microbial dispersal, such as channels angled in zigzag patterns, forcing the microbes to navigate through increasingly sharper turns?"

We added much more detailed information on the chip design via Suppl. Fig 1, and as text to main manuscript in Material and Methods (lines 409-415):

"Section C: A set of channels with sharp corners of three different types (n=12, randomly distributed): zigzag channels (90° turns with all channel sections at ±45° angle from the main growth direction), meandering square channels (90° turns with each section oriented in either the main growth direction or perpendicular to it), z-shaped channels (sharp corners diverting 135° from the previous growth direction with channel sections in the main growth direction and at angles of 45° and 135° from it)"

Line 96-100: English needs rewording as it is not clear at present. Also, the authors should refer to a supplementary figure (see my major comment #2 above) in order to help the reader understand what they mean to convey exactly. It is a little confusing at present.

This comment concerned the approach for analysis of hyphal-bacterial interactions in the chips during Expt 1.

We now explain the chip experimental designs in great detail in the M&M section (lines 395-415), and explain the experimental procedure in detail (lines 498-504): "To analyse the effect of fungal hyphae on bacterial abundance, we recorded real-time videos slowly scanning along the whole length of the diamond-shaped opening channels (each 33 diamonds, section D in Supplementary Fig. 1a; Fig. 3). The rather sparse hyphal colonization allowed us to select pairs of channels where in the first channel a hypha had proliferated far into the channel, combined with a directly adjacent channel without hyphae, n=4. We then counted the number of bacterial cells in diamond-shaped widenings, the presence or absence of fungal hyphae, and the presence or absence of liquid inside each diamond."

Line 101: Why horizontally and not vertically, for example? Please provide a scientific justification for this.

We did not want to introduce extra parameters such as chip inclination.

We now explain in the Methods (lines 476-478): "Horizontal placement was chosen to probe a single stratum of the soil, serving as a comparable inoculum to the whole of the entry system, and to aid non-destructive recovery."

Line 102: Why 64 days? Again, please provide a scientific justification for this time frame.

We now explain in the Methods (lines 483-486): "Preliminary experiments had shown that a two-month incubation period would grant the colonization of different types of soil microorganisms and minerals, and a stabilization of the inner environmental conditions between the soil chip and the surrounding soil. Thus, after 64 days..."

Line 103: "promptly examined under the microscope". What does this mean? Same day? Were the samples stored or the temperature regulated to represent the external

conditions? Perhaps the comparatively warm temperature of the lab cf. the belowground soil floor could influence the observed interactions?

We reformulated this phrase and explain in Material and Methods (lines 492-493): “The chips were harvested one at a time and analysed under the microscope immediately after collection and cleaning. “

We did not regulate the temperature during examination in the laboratory, but due to the prompt analysis we expected this not to have a large influence. After the analysis for data in Figures 1-3, “the chips were left uncovered at room temperature for 60 minutes to initiate air drying in the adjacent soil, in order to observe the real-time effects of drying on organisms and particles in the pore space system of the chips. The adjacent soil was re-wetted by adding 400 µl of water”; lines 505-508.

Line 128-129: The authors mention that there are hypha-free channels directly adjacent to channels containing hyphae. How do they ensure that there are hypha-free channels, which can be directly compared to the hypha-occupying channels? This is unclear, as it is not possible to visualize in which channels these measurements were conducted. Is it just random? Or perhaps the hypha-free channels induce a bias, i.e. exclusion of bacteria due to channel architecture rather than absence of a hypha. This is not clear and needs to be addressed.

Channel architecture was exactly identical. Having 36 channel replicates and a sparse hyphal colonization, we were able to choose this paired design: channels that were adjacent to each other, one with and one without hypha. We now write in lines 500-502: ” The rather sparse hyphal colonization allowed us to select pairs of channels where in the first channel a hypha had proliferated far into the channel, combined with a directly adjacent channel without hyphae, n=4.”

Line 145-146: English needs revising
We removed this sentence.

Line 148: typo
Changed.

Line 158: Do the authors give a reason why fungal hyphae avoided chips with malt agar cf. chips filled with water/air? Do the authors know what fungal species are growing in their chips (perhaps this would give some insight as to why)? This is quite surprising, as other published work show that fungal hyphae grow readily in nutrient-filled microchannels (e.g. Ghanem et al. ES&T, 2019 and Stanley et al. Integr Biol, 2014)

We agree that this was surprising, and after much more experimentation experience we removed this statement. We see a very variable colonization of fungi from soil inocula in the different chip replicates irrespective of chip filling, and random priority effects might play a large role. We now write:

“... fungal hyphae showed variable results, with strong or weak growth without a consistent effect of chip filling in both the field- and lab incubated settings.” (lines 120-121);

We also now discuss more thoroughly in lines 248-252:

“The variation found in our results suggests however that fungal reactions to microspaces can depend on factors not specifically tested in this study, such as priority effects by the order in which individual species entered the channels²⁵, or seasonal variation. It has been suggested that fungal hyphae prefer growing in air-filled pore spaces²⁸, but our study did not confirm this to be generally true.”

Line 161: Please show the data, at least some representative microscopic overviews if possible. This is very interesting and any scientific statements, such as this, should be backed up with data.

This comment is related to the comment above and specifically referred to hyphal colonization of the chips regarding the chip filling, the statement that the strongest hyphal colonization was found in the air-filled chips. After adding several more datasets this statement cannot be clearly supported anymore – we found variable fungal colonization, high or low irrespective of the filling. Therefore, we decided to delete it; see also reasoning above.

Line 400-401: I do not understand what the authors mean with the terminology “water retention bridges”, please explain.

We refer to the viscous water films that can connect solid parts of the soil pore space under semi-dry conditions, and increasingly form under the influence of dissolved organic compounds such as microbial secreted substances (e.g. EPS). We changed this term now to: “Water meniscus connecting soil particles and chip structures”, line 749, which is a term commonly used in soil science.

Supplementary Information: Methods

Line 4: photograph
Changed.

Line 46: specify glass thickness and supplier information

We added to Material and Methods (lines 444-445): “Twelve glass slides sized 55x75 mm and 1 mm thick (Thermo Scientific).…”

Line 68: What is the malt medium exactly?

We added to Material and Methods (lines 457-460): “... liquid malt medium, a complex medium to provide a nutrient-rich environment including reduced sugars such as disaccharide maltose and in lower proportion nitrogenous components such as peptides, amino acids purines and vitamins (malt extract for microbiology, Merck KGaA).…”

Reviewer #3 (Remarks to the Author):

Review of "A window to the underground: microchip gives visual access to in-situ soil ecology"
Mafla-Endara et al.

In this paper, the authors present results related to the microbial colonization of a micro-chip which represents a model for empty spaces in soil. After 2 months of exposure to the natural environment, the micro-chips were recovered and observed mainly with optical microscopy to observe how different microbial groups had invaded its structure. In parallel, experiments of natural community colonization of the chip were conducted in the lab to observe how the dynamics of invasion unfold. In particular, the invasion of an initially-air filled chip with hyphae and the role of these "fungal highways" in leading to subsequent bacteria and protist colonization is the focus of the lab experiments.

By adding the two experiments in the lab, which target specific questions but fit in the overall demonstration of how the chip can be used to investigate ecological questions linked to soil with natural communities, the authors have addressed my main concerns on the previous version of the paper. By downsizing the purely observational parts based on few observations, and introducing instead experiments with large number of replicates, the story becomes much stronger and demonstrate how the chip can be used in the future. Moreover, the novel observation of protists benefiting from the wetness accumulating around hyphae for their own propagation brings a new dimension to the concept of "fungal highways".

I also appreciate the work to make the supplementary videos more readily accessible as I had suggested, with better labelling, etc... They are appealing so it's good to have them more polished.

One should note that the chip design is not so novel anymore, since some of the authors have already published a paper in ISMEJ using this design (<https://doi.org/10.1038/s41396-020-00886-7>). However, the ISMEJ paper focused on specific fungal species, not natural communities, and was purely lab based. Some results published there were also observed in this current version of the paper, e.g. the fact that hyphae generally do not grow far into channels with sharp turns. Still most of the observations of natural communities made in the current paper are distinct from the ISMEJ content.

I have some minor comments below for the authors:

L32: I would say "varying sizes and chemical conditions"

L34: "determinant of soil function and ecological interactions": I suggest to also add the following citation:

Tecon, Ebrahimi, Kleyer, Levi, and Or, "Cell-to-cell bacterial interactions promoted by drier conditions on soil surfaces", PNAS, 115: 9791--9796 (2018)
as an example of changes in ecological interactions driven by change in soil microstructure.

Supplementary Figure 1a: it is great to have the dimensions of the whole chip, however it is unclear what the sizes of the zoomed-in sections A to E are: could we have a scale for each of them too? Alternatively, I would suggest for the authors to overlay in the global view of the chamber small cut-out boxes in dashed lines corresponding exactly to the locations from which the zoomed-in sections A to E were taken. It would help position precisely the sections A to E with respect to the global chip, and also help apprehend their exact dimensions. Color coding of the dashed lines delimiting the regions from which section A to E were taken could help associate each location with the corresponding zoomed-in section.

L117: I find it not very clear what the "resp." stands for. Is it for malt extract and water? Or bacteria and protists? Please clarify. Same comment for L119-120.

L124: the main text mention GIFs 1 to 3, but the Supp Fig 3 mentions GIF 3 to 5. Which one is correct? I think it is GIFs 1 to 3, in which case caption of Fig 3 needs fixing. Apologies for not providing feedback on the GIFs, but I could only access GIF 1 in the submission files.

L132: were we expecting bacteria and protists to have trouble navigating these zigzags? After all, they swim and reorient randomly, can also follow walls... Maybe it would be good to comment on that in the discussion (L251) before moving to the air-filled case.

L133: would it be possible to add "10-um wide channel shapes", to give directly an idea of the geometry to the reader in the result section?

L146: "crevasses" is usually associated with glaciers fractures. Could it be replaced by a more appropriate word?

L156: "...or further behind" In the panel 3f, it seems that the bacteria concentration peaks further behind the tip because the tip of the hyphae itself is in air, so bacteria cannot reach it. If that's the case, then I would suggest to mention it explicitly, for example "or at the end of the water phase closest to the tip" or something similar.

Fig4: Panel c: the font of the tick labels are too small, not readable, please increase.

Line 306: "Our results illustrate that microbes... and the need for anchorage and biofilm formation". This sentence is not very clear, especially the second part "the need..." does not connect well with the first part. Consider clarifying and expanding a bit, for example: "...nutrient supply. This dynamic habitat landscape might promote anchorage and biofilm formation."

Line 362: It's "minuscule" not "miniscule" which is a commonly occurring misspelling.

REVIEWERS' COMMENTS:

Reviewer #1 (Remarks to the Author):

I appreciate the authors' efforts to address my comments. The performance of lab experiments to support observations and address hypotheses from the in situ experiments was a substantial effort and I applaud their efforts. I am satisfied with the revised manuscript and recommend publication.

We thank you for your helpful advice and for showing appreciation of our approach.

Reviewer #2 (Remarks to the Author):

Review Report: A window to the underground – micro-chip gives visual access to in-situ soil ecology Mafla-Endara et al. describe a micro-engineered system that allows researchers to gain visual access to soil ecology in situ. Specifically, the authors placed micro-engineered chips, fabricated from the elastomeric polymer poly(dimethylsiloxane) (PDMS), into soil and left them there for a period of two months. They then dug out the chips and made observations regarding what had colonized the architecture, identifying a variety of soil organisms and processes taking place in the microchannels.

Although very simple in nature, this is – to the best of my knowledge - the first account that details an experimental setup whereby one has put soil onto a micro device, which can then be utilized to investigate soil ecological processes in situ. It is especially interesting that fungal highways have been observed in situ using real soils. This study highlights how lab-on-a-chip technology can help to elucidate (and begin to quantify) processes taking place in soil environments. It is refreshing to see that the authors also appreciate the limitations of the device setup, whilst at the same time informing of the potential future advantages.

We thank you for showing appreciation of our approach.

The authors have addressed all of my comments adequately; however, I have a very minor additional request for Point 2:

I appreciate that the authors have made a very good effort to improve the level of detail regarding the chip design. However, some information is still lacking and needs clarifying. Section C: Please include microchannel width here too (so that readers have all information easily accessible in one place). It is also not possible to determine how long the channels are before the channel changes direction from this drawing.

We expanded the caption text with a detailed description of the structures. Now it is written:

“Supplementary Figure 1. The soil chip and its use in soil. a, Overview over the design of the chip’s internal structures, which contains five particular experimental sections, A-E as depicted in the top, and with detailed, zoomed-in illustrations for each section below and to the right. The red squares indicate the position of the high-magnification illustrations in the chip overview for each

respective experimental section. Section A represents blocks of hexagonal pillars with different diameters thus creating different porosities (n=24). Section B consists of straight channels of different widths (20, 15, 10, 8, 6, 4 μm) with five replicates of each type. Section C corresponds to a set of 10 μm wide angled channels with three different types of turning angle/arrangement (zigzag, square and z-shaped, n=12) The segments in between the angles in zigzag and square design are 100 μm ; the long segments of the z-shaped channels are 200 μm . The total length of the zigzag channels is 27000 μm , the other channel types were analyzed to the same length only. Section D consists of repeated patterns of 10 μm width straight channels interspersed by widenings of a diameter of 140 μm . Section E includes two different obstacle courses containing complex structures in irregular arrangement. Scale bar = 100 μm .”

Section E: This is still far too small

It is true that the details of the Section E, consisting of several different complex “obstacle courses” are not easily visible, but our attempt is to provide an overview of the variation of the used shapes, rather than the exact shapes in detail, as they are not used for any specific questions in this study but only were screened for general observations. We chose therefore to retain the overview illustration instead of changing it to a higher zoom on fewer details – but can of course change this if you still consider it.

A scale bar needs to be included with each of the enlarged sections A-E.

We added a scale bar and we also overlaid the overview of the design with red boxes that correspond to the size-exact locations of the zoomed-in sections A-E.

In light of the new material that has been included in the manuscript, I have a few additional minor comments:

- The movies have been greatly improved; I really like them now. However, I was a little confused at first though by the key/legend, as I was expecting hyphae etc to be present in Movie 1 for example. As a suggestion, I think it would be more reader-friendly to include only those elements in the key/legend that are displayed in the cartoon overview (e.g. chip, soil, air, water for Movie 1).

We are glad you like the new version of the movies. We have now changed the legends to match exactly as the elements seen in the movie.

- Please include the cartoon overviews in all movies (only Movies 1-8 have this).

Following your suggestion, we included the cartoon overviews in all movies except for supplementary movies 11 and 12 (particle tracking) where there is no need.

- Is Movie 4 (Habitat fragmentation) actually a movie? I do not see anything happening (at least it is not obvious viewing it on my monitor).

The bacterial cells are actually moving around the fungal hyphae, especially visible as moving shadows in the left and right corners of the diamond-shaped pore. It is unfortunate that it cannot be seen in all monitors – try increasing the size of the video window, enlarging helps.

- Each GIF needs to include a scale bar and a title (so that it is easy for the reader to grasp / remind them what it represents). The GIFs are a nice idea, however it is difficult to check both the time point and what is changing in the channels from frame to frame. Please include a Supplementary Figure for each GIF showing a time series (essentially each frame in the GIF side by side in a figure) with a description of what is happening. A description in the main manuscript is lacking and it is therefore not clear what each GIF is depicting.

Following your recommendation, we now added a scale bar and a title. In addition, we included supplementary figures (Suppl. Figs. 3, 4) with the sequences of the images and a description.

- Fig 3C: please label top axis. Does this refer to the percentage of channels that either contain or do not contain hyphae?

We added a label for the top axis which corresponds to the percentage of channels.

- Line 804: revise English “was kept equable”

Changed to “kept equal” (line 845).

- Line 221: “by breaking up solid structures”. Please rephrase this sentence. Sometimes PDMS does not always bond properly/uniformly to glass. Therefore, in such regions you will see the hyphae squeezing between the PDMS and glass layers. I very much doubt that the hyphae are breaking the covalent bonds formed between the glass and PDMS during plasma bonding (unless you can prove this?) and you would have major problems everywhere in your device if this was the case.

Changed to “squeezing between the PDMS and the glass of the chips” (line 215).

- I find it very confusing as to what n refers to exactly, i.e. when the authors describe each Experiment (lines 465 onwards). For example, in line 466 – 467 three filling treatments are described (deionised water, malt extract, air), and ‘n’ is said to equal 3. What is n here: 1 replicate for each condition, or 3 replicates of each condition. This needs to be clarified throughout the manuscript.

Statistical replication should of course always refer to the number of treatments/conditions to be compared, and we now make this very clear and specifically write e.g. “n=3 chips per treatment” (line 475, line 479); “n= 2 chips x 12 channels” (line 121), etc.

In summary, I applaud the authors on improving their study greatly with the addition of the new experiments and look forward to seeing this manuscript published in Communications Biology.

Thank you, very nice to hear you like it.

Reviewer #3 (Remarks to the Author):

Review of "A window to the underground: microchip gives visual access to in-situ soil ecology" Mafla-Endara et al.

In this paper, the authors present results related to the microbial colonization of a micro-chip which represents a model for empty spaces in soil. After 2 months of exposure to the natural environment, the micro-chips were recovered and observed mainly with optical microscopy to observe how different microbial groups had invaded its structure. In parallel, experiments of natural community colonization of the chip were conducted in the lab to observe how the dynamics of invasion unfold. In particular, the invasion of an initially-air filled chip with hyphae and the role of these "fungal highways" in leading to subsequent bacteria and protist colonization is the focus of the lab experiments.

By adding the two experiments in the lab, which target specific questions but fit in the overall demonstration of how the chip can be used to investigate ecological questions linked to soil with natural communities, the authors have addressed my main concerns on the previous version of the paper. By downsizing the purely observational parts based on few observations, and introducing instead experiments with large number of replicates, the story becomes much stronger and demonstrate how the chip can be used in the future. Moreover, the novel observation of protists benefiting from the wetness accumulating around hyphae for their own propagation brings a new dimension to the concept of "fungal highways".

I also appreciate the work to make the supplementary videos more readily accessible as I had suggested, with better labelling, etc... They are appealing so it's good to have them more polished.

Thanks for showing your appreciation.

One should note that the chip design is not so novel anymore, since some of the authors have already published a paper in ISMEJ using this design (<https://doi.org/10.1038/s41396-020-00886-7>). However, the ISMEJ paper focused on specific fungal species, not natural communities, and was purely lab based. Some results published there were also observed in this current version of the paper, e.g. the fact that hyphae generally do not grow far into channels with sharp turns. Still most of the observations of natural communities made in the current paper are distinct from the ISMEJ content.

Correct, the chip was not newly designed for this study and was already used in the ISMEJ paper, to which we refer at first mention of the chip in the end of the

Introduction (line 81-82) and the beginning of the Methods section, now stating even more explicitly:

"We used our micro-engineered silicone chip termed the "Obstacle chip"²⁶..." (line 404)

However, as you say, the scientific questions addressed are very different and novel, glad you agree on this.

I have some minor comments below for the authors:

L32: I would say "varying sizes and chemical conditions"

Agreed and changed (line 32).

L34: "determinant of soil function and ecological interactions": I suggest to also add the following citation:

Tecon, Ebrahimi, Kleyer, Levi, and Or, "Cell-to-cell bacterial interactions promoted by drier conditions on soil surfaces", PNAS, 115: 9791--9796 (2018)

as an example of changes in ecological interactions driven by change in soil microstructure.

We added this reference (line 34).

Supplementary Figure 1a: it is great to have the dimensions of the whole chip, however it is unclear what the sizes of the zoomed-in sections A to E are: could we have a scale for each of them too? Alternatively, I would suggest for the authors to overlay in the global view of the chamber small cut-out boxes in dashed lines corresponding exactly to the locations from which the zoomed-in sections A to E were taken. It would help position precisely the sections A to E with respect to the global chip, and also help apprehend their exact dimensions. Color coding of the dashed lines delimiting the regions from which section A to E were taken could help associate each location with the corresponding zoomed-in section.

Following your advice, we overlaid the overview of the design with red boxes that correspond to the locations of the zoomed-in sections A-E, and we added a scale bar. Furthermore, we expanded the caption text with a detailed description of the structures. Now it is written:

"Supplementary Figure 1. The soil chip and its use in soil. a, Overview over the design of the chip's internal structures, which contains five particular experimental sections, A-E as depicted in the top, and with detailed, zoomed-in illustrations for each section below and to the right. The red squares indicate the position of the high-magnification illustrations in the chip overview for each respective experimental section. Section A represents blocks of hexagonal pillars with different diameters thus creating different porosities (n=24). Section B consists of straight channels of different widths (20, 15, 10, 8, 6, 4 μm) with five replicates of each type. Section C corresponds to a set of 10 μm wide angled channels with three different types of turning angle/arrangement (zigzag, square and z-shaped, n=12)

The segments in between the angles in zigzag and square design are 100 μm ; the long segments of the z-shaped channels are 200 μm . The total length of the zigzag channels is 27000 μm , the other channel types were analyzed to the same length only. Section D consists of repeated patterns of 10 μm width straight channels interspersed by widenings of a diameter of 140 μm . Section E includes two different obstacle courses containing complex structures in irregular arrangement. Scale bar = 100 μm ."

L117: I find it not very clear what the "resp." stands for. Is it for malt extract and water? Or bacteria and protists? Please clarify. Same comment for L119-120.

We changed the text to the following for improved clarity: **Fig. 2b, c**; $F= 4.0$, $p=0.037$ for bacteria and $F= 3.63$, $p=0.047$ for protists; $n=3$, $DF=8$), (line 118-119).

L124: the main text mention GIFs 1 to 3, but the Supp Fig 3 mentions GIF 3 to 5. Which one is correct? I think it is GIFs 1 to 3, in which case caption of Fig 3 needs fixing. Apologies for not providing feedback on the GIFs, but I could only access GIF 1 in the submission files.

Thank you for noticing this, we now changed the GIF numbering to correct order, 1-3.

L132: were we expecting bacteria and protists to have trouble navigating these zigzags? After all, they swim and reorient randomly, can also follow walls... Maybe it would be good to comment on that in the discussion (L251) before moving to the air-filled case.

Agree, we now added the following sentence to the discussion (line 254-256): "The water-dwelling organism groups bacteria and protists were, as expected, not affected in their principle dispersal capabilities by the channel geometry".

L133: would it be possible to add "10-um wide channel shapes", to give directly an idea of the geometry to the reader in the result section?

Added (line 132).

L146: "crevasses" is usually associated with glaciers fractures. Could it be replaced by a more appropriate word?

We changed "crevasses" to "pore structures" (line 148).

L156: "...or further behind" In the panel 3f, it seems that the bacteria concentration peaks further behind the tip because the tip of the hyphae itself is in air, so bacteria cannot reach it. If that's the case, then I would suggest to mention it explicitly, for example "or at the end of the water phase closest to the tip" or something similar.

We now write: "Bacterial cell abundance commonly peaked near the hyphal tip (**Fig. 3e**), or until where towards the furthest extent the water reached along the hypha (**Fig. 3f**)" (line 157-159).

Fig4: Panel c: the font of the tick labels are too small, not readable, please increase.

We have now changed it to a more readable size.

Line 306: "Our results illustrate that microbes... and the need for anchorage and biofilm formation". This sentence is not very clear, especially the second part "the need..." does not connect well with the first part. Consider clarifying and expanding a bit, for example: "...nutrient supply. This dynamic habitat landscape might promote anchorage and biofilm formation."

Agreed, we changed according to your suggestion (line 314).

Line 362: It's "minuscule" not "miniscule" which is a commonly occurring misspelling.

Changed, thanks for noticing this.